# Structure and function of the Smoothened extracellular domain in vertebrate Hedgehog signaling

Sigrid Nachtergaele[1†], Daniel M Whalen[2†], Laurel K Mydock[3], Zhonghua Zhao[4], Tomas Malinauskas[2], Kathiresan Krishnan[3], Philip W Ingham[4,5], Douglas F Covey[3], Christian Siebold[2*], Rajat Rohatgi[1,6*]

[1]Department of Biochemistry, Stanford University School of Medicine, Stanford, United States; [2]Division of Structural Biology, Wellcome Trust Centre for Human Genetics, University of Oxford, Oxford, United Kingdom; [3]Department of Developmental Biology, Washington University School of Medicine, St. Louis, United States; [4]A*STAR Institute of Molecular and Cell Biology, Singapore, Singapore; [5]Lee Kong Chian School of Medicine, Imperial College London/Nanyang Technological University, Singapore, Singapore; [6]Department of Medicine, Stanford University School of Medicine, Stanford, United States

*For correspondence: christian@strubi.ox.ac.uk (CS); rrohatgi@stanford.edu (RR)

[†]These authors contributed equally to this work

**Abstract** The Hedgehog (Hh) signal is transduced across the membrane by the heptahelical protein Smoothened (Smo), a developmental regulator, oncoprotein and drug target in oncology. We present the 2.3 Å crystal structure of the extracellular cysteine rich domain (CRD) of vertebrate Smo and show that it binds to oxysterols, endogenous lipids that activate Hh signaling. The oxysterol-binding groove in the Smo CRD is analogous to that used by Frizzled 8 to bind to the palmitoleyl group of Wnt ligands and to similar pockets used by other Frizzled-like CRDs to bind hydrophobic ligands. The CRD is required for signaling in response to native Hh ligands, showing that it is an important regulatory module for Smo activation. Indeed, targeting of the Smo CRD by oxysterol-inspired small molecules can block signaling by all known classes of Hh activators and by clinically relevant Smo mutants.

## Introduction

The Hedgehog (Hh) signaling pathway controls the development of many tissues during embryogenesis (*McMahon et al., 2003*). Even quantitative abnormalities in Hh signaling can lead to human birth defects (*Bale, 2002*). After development, Hh signaling regulates tissue stem cells and regenerative responses to injury (*Machold et al., 2003*; *Shin et al., 2011*). Aberrant Hh signaling can be oncogenic, and genes encoding Hh pathway proteins can function as oncogenes or tumor suppressor genes (*Scales and de Sauvage, 2009*). The most commonly damaged step in Hh-driven cancers involves the poorly understood interaction between two transmembrane (TM) proteins, Patched 1 (Ptch1) and Smoothened (Smo) (reviewed in *Briscoe and Therond [2013]*). Ptch1, encoded by a tumor suppressor gene, is a 12-pass TM protein that serves as the receptor for Hh ligands, including Sonic Hedgehog (Shh) (*Marigo et al., 1996*; *Stone et al., 1996*). In the absence of Hh ligands, Ptch1 inhibits the function of Smo, a 7-pass TM protein that is encoded by a human oncogene. Shh binds and inactivates Ptch1, unleashing Smo's activity and allowing the Gli transcription factors to initiate target gene transcription. Despite the fact that Smo has become a drug target in oncology, with an FDA-approved Smo inhibitor in clinical use (*Von Hoff et al., 2009*) and others in ongoing trials, the mechanism by which Smo is regulated by Ptch1 remains a mystery. Current models suggest that Ptch1, a protein with some homology

**eLife digest** Just over 30 years ago, researchers identified a new signaling molecule with an important role in the development of fruit flies. Embryos lacking this molecule were thought to resemble a hedgehog, eventually leading to this cell–cell communication system being designated the "Hedgehog" pathway. This pathway has subsequently been shown to be involved in the development of many other animals, as well as in the repair of damaged tissues in adult organisms.

Abnormal Hedgehog signaling has also been implicated in both human birth defects and in cancers of the skin and the brain. Many such tumors are driven by the unrestrained activation of a membrane-bound protein called Smoothened, which has led to the development and clinical use of small molecules that prevent Hedgehog from activating Smoothened. The existing anti-tumor drugs all bind to the same region of the Smoothened receptor, namely the part that sits within the cell membrane. A second group of molecules, known as oxysterols, can activate Smoothened, but exactly how they do this has been unclear. Now, Nachtergaele et al. have shown that oxysterols bind to a region of the Smoothened receptor that lies outside the cell, and that is rich in the amino acid cysteine.

By solving the crystal structure of this part of the receptor from zebrafish, Nachtergaele et al. were able to map the oxysterol binding site at high resolution. This revealed strong similarities between this binding site and those in related receptors belonging to the Wnt signaling pathway. Deleting the cysteine-rich domain significantly impaired Hedgehog signaling, as did a new class of small molecule inhibitors designed specifically to target the oxysterol binding site.

In addition to providing new insights into the structure and function of the Smoothened receptor, the work of Nachtergaele et al. opens up possibilities for novel therapeutic agents that could be used in the treatment of cancers caused by abnormal Hedgehog signaling.

to bacterial small molecule transporters, regulates Smo through an endogenous ligand whose identity is unknown (*Davies et al., 2000*; *Taipale et al., 2002*).

Smo consists of an extracellular N-terminal region containing a cysteine rich domain (CRD), a heptahelical transmembrane segment (7TM) and an intracellular C-terminal tail (C-term) (*Figure 1A*). Smo belongs to the G-protein coupled receptor (GPCR) superfamily of proteins, most closely related to the Frizzled (Fz) group of Wnt receptors (*Dann et al., 2001*; *Fredriksson et al., 2003*). Previous work on Smo has largely focused on the 7TM domain, which contains a binding site for cyclopamine, a sterol-like plant alkaloid that was the foundational Hh inhibitor (*Chen et al., 2002a*). A battery of subsequent small-molecule screens uncovered a set of exogenous ligands that regulate Smo activity through this site, either as agonists such as SAG or antagonists such as SANT-1 and the FDA-approved Hh-inhibitor Vismodegib (*Frank-Kamenetsky et al., 2002*; *Chen et al., 2002b*; *Robarge et al., 2009*). The 2.5 Å crystal structure of the 7TM segment of Smo bound to a synthetic antagonist has provided a high-resolution view of this binding pocket, which is formed by the extracellular end of the 7TM helix bundle and connecting loops (*Wang et al., 2013*). Smo drugs that occupy this 'cyclopamine binding site' are classified as such by their ability to compete with cyclopamine for Smo binding. No endogenous molecules are known that engage this site in the 7TM of Smo.

A second binding site on Smo has been defined by side-chain oxysterols, oxidized derivatives of cholesterol carrying an additional hydroxyl group on the *iso*-octyl chain. Specific oxysterols can fully activate Hh signaling in the absence of Hh ligands in multiple cell types and also induce the accumulation of Smo in the primary cilium, a trafficking step essential for Smo to activate downstream signaling (*Kha et al., 2004*; *Corcoran and Scott, 2006*; *Dwyer et al., 2007*; *Kim et al., 2007*; *Rohatgi et al., 2007*; *Johnson et al., 2011*). We previously demonstrated that a specific side-chain oxysterol, 20(*S*)-hydroxycholesterol (20(*S*)-OHC), directly binds Smo in a manner that is highly stereospecific: the enantiomer, *ent*-20(*S*)-OHC, or the epimer, 20(*R*)-OHC, failed to bind Smo or to activate Hh signaling (*Nachtergaele et al., 2012*). While this 'oxysterol binding site' showed allosteric interactions with the canonical cyclopamine binding site, it was clearly distinct since oxysterols did not show a competitive interaction with cyclopamine (*Dwyer et al., 2007*; *Nachtergaele et al., 2012*). Indeed, previous structural comparison studies have speculated that oxysterols bind to the extracellular CRD of Smo based on its relationship to the Fz CRD, which binds to the palmitoleyl group of secreted Wnt ligands (*Bazan and de Sauvage, 2009*; *Bazan et al., 2012*; *Janda et al., 2012*; *Sharpe and de Sauvage, 2012*).

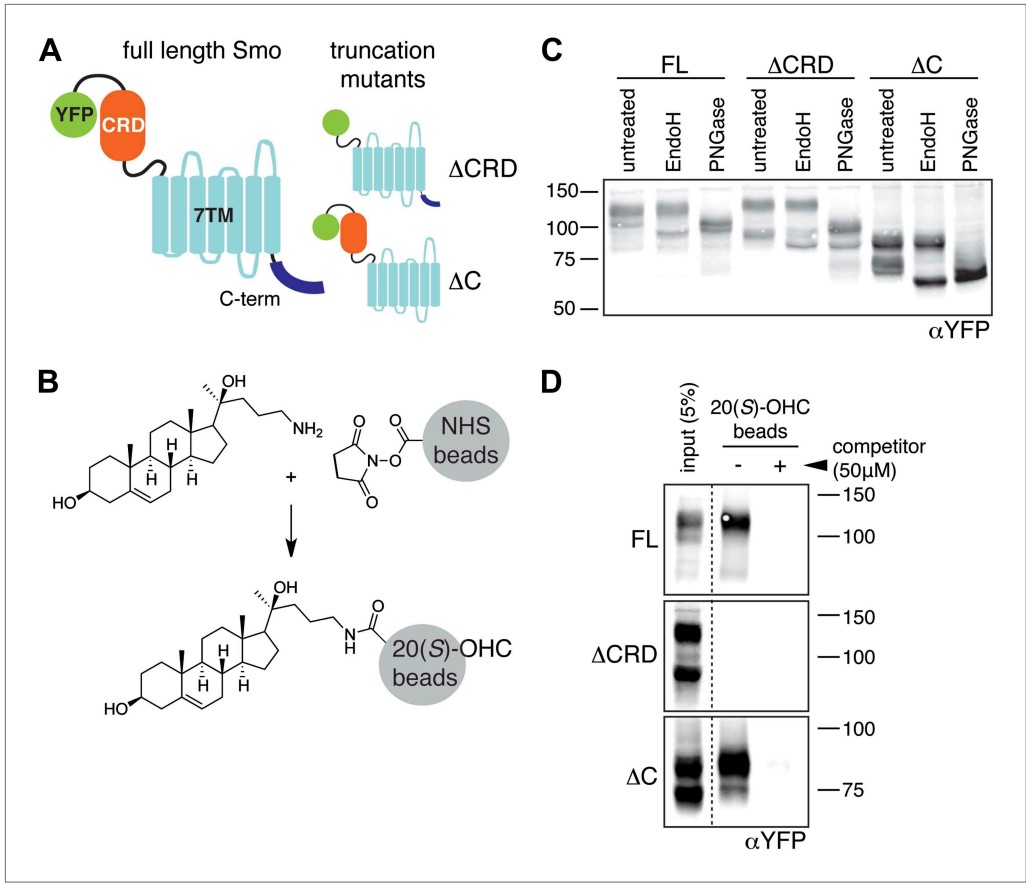

**Figure 1**. The mouse Smo CRD is required to bind oxysterols. (**A**) Schematic of full-length (FL), YFP-tagged mSmo and the ΔCRD and ΔC truncation mutants used in this study. (**B**) Structure of the 20(*S*)-OHC beads used in Smo pull-down assays. (**C**) EndoH and PGNaseF sensitivity of YFP-mSmo, ΔCRD-YFP-mSmo and ΔC-YFP-mSmo stably expressed in *Smo⁻/⁻* cells and loaded on an 8% Tris-Glycine SDS-PAGE gel. The fraction of each protein with slower mobility on the gel was resistant to EndoH but sensitive to PGNaseF, suggesting post-Golgi localization. (**D**) 20(*S*)-OHC beads captured YFP-mSmo and ΔC-YFP-mSmo, but not ΔCRD-YFP-mSmo from lysates of cells stably expressing each protein. Binding to beads was not seen when 50 μM free 20(*S*)-OHC was added as a competitor.

Wnt binding to the Fz CRD triggers signaling across the membrane, but the function of the Smo CRD has remained enigmatic.

We find that the extracellular CRD of Smo in vertebrates is both necessary and sufficient to bind to 20(*S*)-OHC, thus demonstrating that the cyclopamine and oxysterol binding sites occupy different domains in Smo. We determined the crystal structure of the zebrafish Smo CRD at 2.3 Å to provide a view of the oxysterol-binding pocket and to establish its relationship to the Fz CRD and other Fz-like CRDs that bind small hydrophobic ligands. Either deletion of the CRD or its inhibition by a new class of oxysterol-inspired small molecules can impair the signaling initiated by the native ligand Shh. Our results elucidate the molecular mechanism by which oxysterols activate Smo and show that the Smo CRD is a physiologically and therapeutically important target in the vertebrate Hh pathway.

## Results

### The extracellular CRD of mouse Smo is necessary and sufficient for binding to 20(*S*)-OHC

We previously developed a ligand affinity chromatography assay to measure the interaction between 20(*S*)-OHC and detergent-solubilized, full-length Smo (**Nachtergaele et al., 2012**). For the studies presented here, we used a similar strategy to assay the interaction between truncated versions of Smo (**Figure 1A**) and 20(*S*)-OHC, using sepharose beads on which 20(*S*)-OHC was immobilized through an

amino group installed on the *iso*-octyl chain (hereafter called 20(*S*)-OHC beads; *Figure 1B*). We produced deletion mutants (*Figure 1A*) of yellow fluorescent protein (YFP)-tagged mouse Smo (mSmo) lacking the CRD (ΔCRD-YFP-mSmo) or the C-terminal intracellular domain (ΔC-YFP-mSmo) and confirmed that these proteins were folded when stably expressed in Smo$^{-/-}$ mouse embryonic fibroblasts (MEFs) (*Rohatgi et al., 2009*). Both truncated proteins demonstrated a slower migrating species that was resistant to Endoglycosidase H (EndoH), suggesting the presence of glycan modifications usually attached in the Golgi (*Figure 1C*) (*Chen et al., 2002a*). For both YFP-mSmo and ΔC-YFP-mSmo, this post-Golgi band was selectively captured on 20(*S*)-OHC beads, showing that the C-terminal intracellular domain of Smo was dispensable for this interaction (*Figure 1D*). In this and subsequent experiments, specificity of binding was established by competition with free 20(*S*)-OHC. In contrast, ΔCRD-YFP-mSmo failed to show an interaction, suggesting that the CRD was required for oxysterol binding. Previous studies have shown that the truncated versions of Smo lacking either CRD or the C-terminal domain remain competent to bind cyclopamine and other cyclopamine-competitive ligands, consistent with these molecules interacting with the 7TM segment (*Chen et al., 2002a*; *Wang et al., 2013*). ΔCRD-YFP-mSmo also remained responsive to 7TM ligands (described below), confirming proper folding.

To determine if the mSmo CRD was sufficient to bind 20(*S*)-OHC, we purified isolated mSmo CRD fused to the constant region of the human IgG heavy chain (mSmo CRD-Fc; *Figure 2A*). The mSmo CRD-Fc protein secreted into the media of 293F cells ran as a smear on an SDS-PAGE gel. Further purification by Protein A affinity chromatography followed by gel filtration allowed us to isolate mono-disperse mSmo CRD-Fc (*Figure 2A*, fractions 13–15). This well-behaved protein bound to 20(*S*)-OHC beads. A significant population of the protein was clearly misfolded, as it fractionated as a broad peak on a gel filtration column and failed to bind to 20(*S*)-OHC (*Figure 2A*, fractions 5–12). Binding of mSmo CRD-Fc to 20(*S*)-OHC beads was saturable (*Figure 2B*), specific (*Figure 2C*) and followed the same requirements for oxysterol stereochemistry and regiochemistry as previously described (*Figure 2D*) (*Nachtergaele et al., 2012*). Binding could be inhibited by free 20(*S*)-OHC and free 20(*S*)-yne, the ~10-fold more potent alkyne analog of 20(*S*)-OHC (*Nachtergaele et al., 2012*). However, the enantiomer *ent*-20(*S*)-OHC, the epimer 20(*R*)-OHC, and 22(*S*)-OHC (all sterols that cannot activate Hh signaling) were unable to inhibit binding (*Nachtergaele et al., 2012*). Ligands known to engage Smo at the cyclopamine binding site, SAG and SANT-1, failed to inhibit the binding of mSmo CRD-Fc to 20(*S*)-OHC beads, as did Itraconazole, a purported Smo ligand that binds to an unknown site (*Figure 2E*) (*Chen et al., 2002a*, *2002b*; *Kim et al., 2010*). While our manuscript was in preparation, an independent study also reported the interaction between oxysterols and the Smo CRD (*Nedelcu et al., 2013*). Overall, our results show that the cyclopamine and oxysterol binding sites on Smo are distinct. For clarity, we hereafter refer to these sites as the 7TM and CRD sites, respectively.

## The Smo CRD is required for Shh-induced signaling

To investigate the function of the Smo CRD for signaling induced by the native ligand Shh, YFP-tagged mSmo variants were expressed by stable retroviral transduction in Smo$^{-/-}$ MEFs to avoid the confounding effects of endogenous Smo. These clonal Smo$^{-/-}$:YFP-mSmo cells could activate a Hh target gene, Gli1, when exposed to Shh (which binds and inactivates Ptch1) or to the Smo agonists SAG and 20(*S*)-OHC, which bind to the 7TM and CRD sites, respectively (*Figure 3A*) (*Rohatgi et al., 2009*). In an independent, non-transcriptional measure of signaling, loss of the repressor form of Gli3 (Gli3R) was observed in response to all three agonists (*Figure 3A*). In contrast, ΔCRD-YFP-mSmo failed to activate Hh target genes or to extinguish Gli3R levels in response to both Shh and 20(*S*)-OHC, but retained its ability to respond to SAG (*Figure 3A*). Identical results were obtained using a luciferase-based Hh reporter transiently expressed along with YFP-mSmo or ΔCRD-YFP-mSmo in Smo$^{-/-}$ cells (*Figure 3B*) (*Sasaki et al., 1997*; *Varjosalo et al., 2006*). SAG activated ΔCRD-YFP-mSmo remained susceptible to inhibition by cyclopamine, consistent with an intact 7TM site (*Figure 3C*). As noted previously, ΔCRD-YFP-mSmo was not constitutively active, but it did demonstrate a higher level of basal activity in Hh reporter assays (*Taipale et al., 2002*; *Aanstad et al., 2009*). The SAG responsiveness shows that ΔCRD-YFP-mSmo is not a misfolded or inactive protein; instead, it supports the notion that the CRD of Smo mediates the response to oxysterols while the 7TM segment mediates the response to SAG. Most significantly, this result suggests that the CRD plays an important role in mediating the response to Shh and thus in mediating the interaction between Ptch1 and Smo.

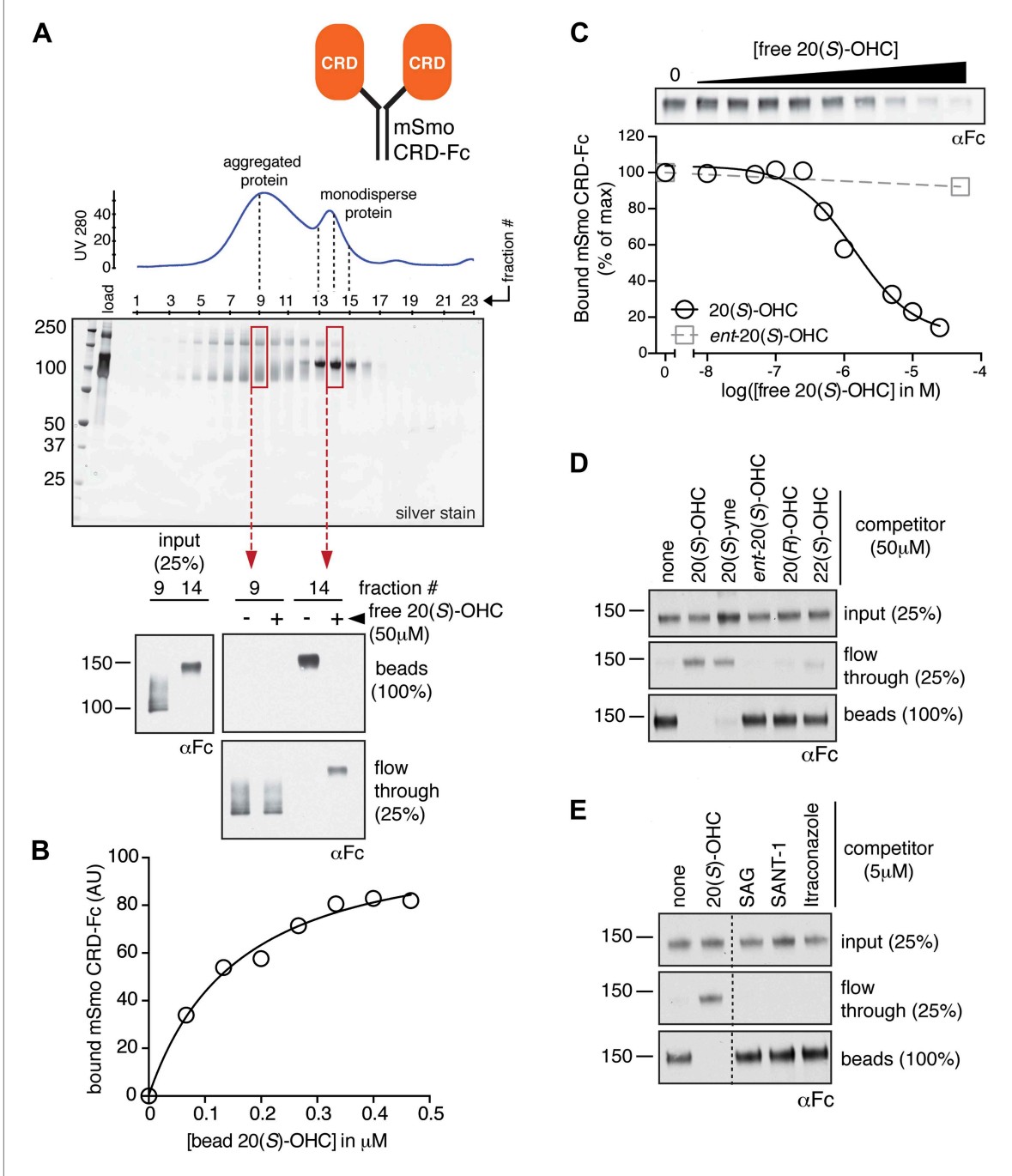

**Figure 2**. The isolated mSmo CRD can bind oxysterols. (**A**) Fractionation of the mSmo CRD-Fc protein on a Superose 6 gel-filtration column. The UV280 absorbance of each fraction (blue curve) is shown above the protein content of each fraction on a silver stained gel. Monodisperse protein (fractions 13–15) elutes in a sharp peak and binds to 20(*S*)-OHC beads (panels below), while aggregated protein runs as a broad peak (fractions 5–12) and fails to bind oxysterols. The indicated fractions (red boxes) were incubated with 20(*S*)-OHC beads in the presence or absence of free 20(*S*)-OHC competitor, and the amount of mSmo CRD-Fc protein captured on the beads or left in the flow through was assayed on an anti-Fc immunoblot. (**B**) A binding curve ($K_d$ ~180 nM) for the mSmo CRD-Fc-20(*S*)-OHC interaction was measured by incubating a fixed amount of protein with increasing amounts of bead-immobilized sterol. (**C**) Binding of mSmo CRD-Fc to 20(*S*)-OHC beads is inhibited in a dose-responsive fashion by free 20(*S*)-OHC but not by the enantiomer *ent*-20(*S*)-OHC. A competition assay was used to test the ability of various oxysterols (**D**) or Smo ligands (**E**) to inhibit the binding of mSmo CRD-Fc to 20(*S*)-OHC beads. Anti-Fc immunoblots show the amount of protein in the input, captured on the beads, and left in the flow-through.

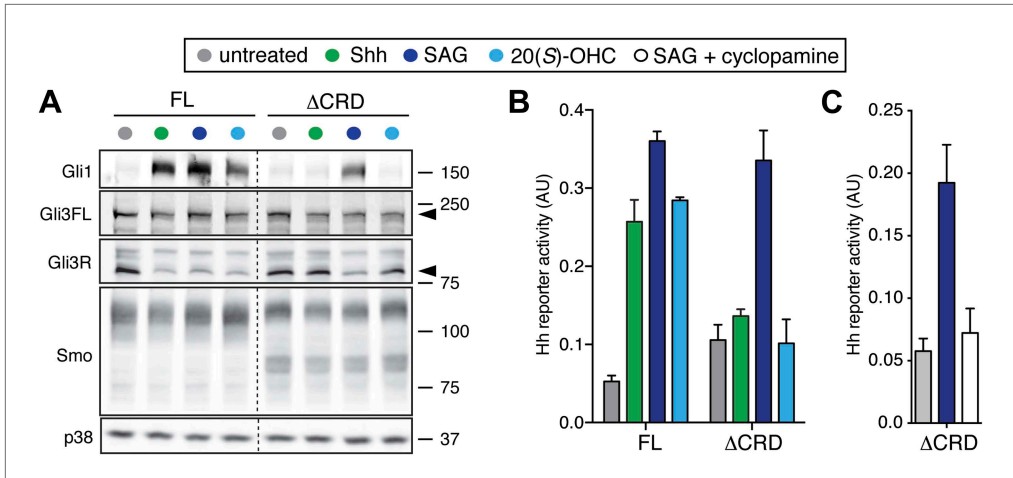

**Figure 3**. The mSmo CRD is required for Shh- and oxysterol-mediated activation of Hh signaling. (**A**) Smo$^{-/-}$ cells stably expressing full-length (FL) YFP-mSmo or ΔCRD-YFP-mSmo were treated with Shh, SAG (100 nM) or 20(*S*)-OHC (10 μM). Levels of Gli1 and Gli3R protein, determined by immunoblotting after fractionation on an 8% Tris-glycine SDS-PAGE gel, were taken as a metric of pathway activation. An anti-YFP blot shows the levels of YFP-mSmo in each sample, and p38 levels are used as a loading control. (**B** and **C**) A luciferase-based Hh reporter gene was used to measure signaling in Smo$^{-/-}$ cells transiently transfected with constructs encoding YFP-mSmo or ΔCRD-YFP-mSmo and then treated with the indicated Smo ligands. In (**C**), ΔCRD-YFP-mSmo activated with SAG (25 nM) can be inhibited by the co-administration of cyclopamine (5 μM). Error bars denote S.D. (n = 3).

## The oxysterol–CRD interaction is conserved across vertebrates

We tested the binding of Smo from various species to 20(*S*)-OHC beads (**Figure 4A,B**). Both full-length *Drosophila melanogaster* Smo (dSmo) and the isolated dSmo CRD failed to bind 20(*S*)-OHC beads. However, a truncated version of zebrafish Smo (zSmo) lacking the intracellular C-terminal region (YFP-zSmoΔC), expressed in mammalian cells and solubilized with detergent, bound to 20(*S*)-OHC beads, showing that this interaction is likely conserved in the vertebrate (but not in the invertebrate) Hh pathway.

We tested whether the zebrafish Hh pathway was responsive to oxysterols, because our structural studies described below focused on Smo protein from this species. Full-length zebrafish Smo was poorly expressed in mammalian cells, precluding tests of its responsiveness to oxysterols in cultured cells. Hh pathway activity underlies the specification of distinct muscle cell types in the zebrafish embryo, in part through the activation of the *engrailed2* (*eng2*) gene in subsets of slow-twitch and fast-twitch myoblasts (**Wolff et al., 2003**). To investigate the in vivo significance of the interaction between 20(*S*)-OHC and Smo, we treated embryos carrying an eng2a:GFP reporter construct (**Maurya et al., 2011**) with either 20(*S*)-OHC or cyclopamine. As expected, cyclopamine treatment suppressed expression of the reporter gene; by contrast, 20(*S*)-OHC treated embryos showed a significant increase in the number of GFP positive fast twitch muscles compared to vehicle treated embryos (**Figure 4C** and **Figure 4—figure supplement 1**). Consistent with this, 20(*S*)-OHC treated embryos also showed a modest expansion of the expression domain of the endogenous Hh target gene, *ptch2*. These data show that 20(*S*)-OHC can induce Hh signaling in the context of a living vertebrate embryo and suggest that the in vitro interaction between zebrafish Smo and 20(*S*)-OHC induces its activation in vivo.

We succeeded in purifying large quantities of the zSmo ectodomain, encompassing both the CRD and the segment between the CRD and the first transmembrane helix. The zSmo ectodomain demonstrated saturable, specific binding to 20(*S*)-OHC beads (**Figure 4D,E**). Similar to the mouse protein, binding could be inhibited by oxysterols that activate Hh signaling but not by those that cannot (**Figure 4F**); 7TM site ligands also failed to compete for binding (**Figure 4G**).

## The structure of the Smo CRD from zebrafish

To obtain molecular insights into the architecture of the Smo extracellular region, we crystallized the zSmo ectodomain and determined its structure using selenomethionine-labeled protein for phasing

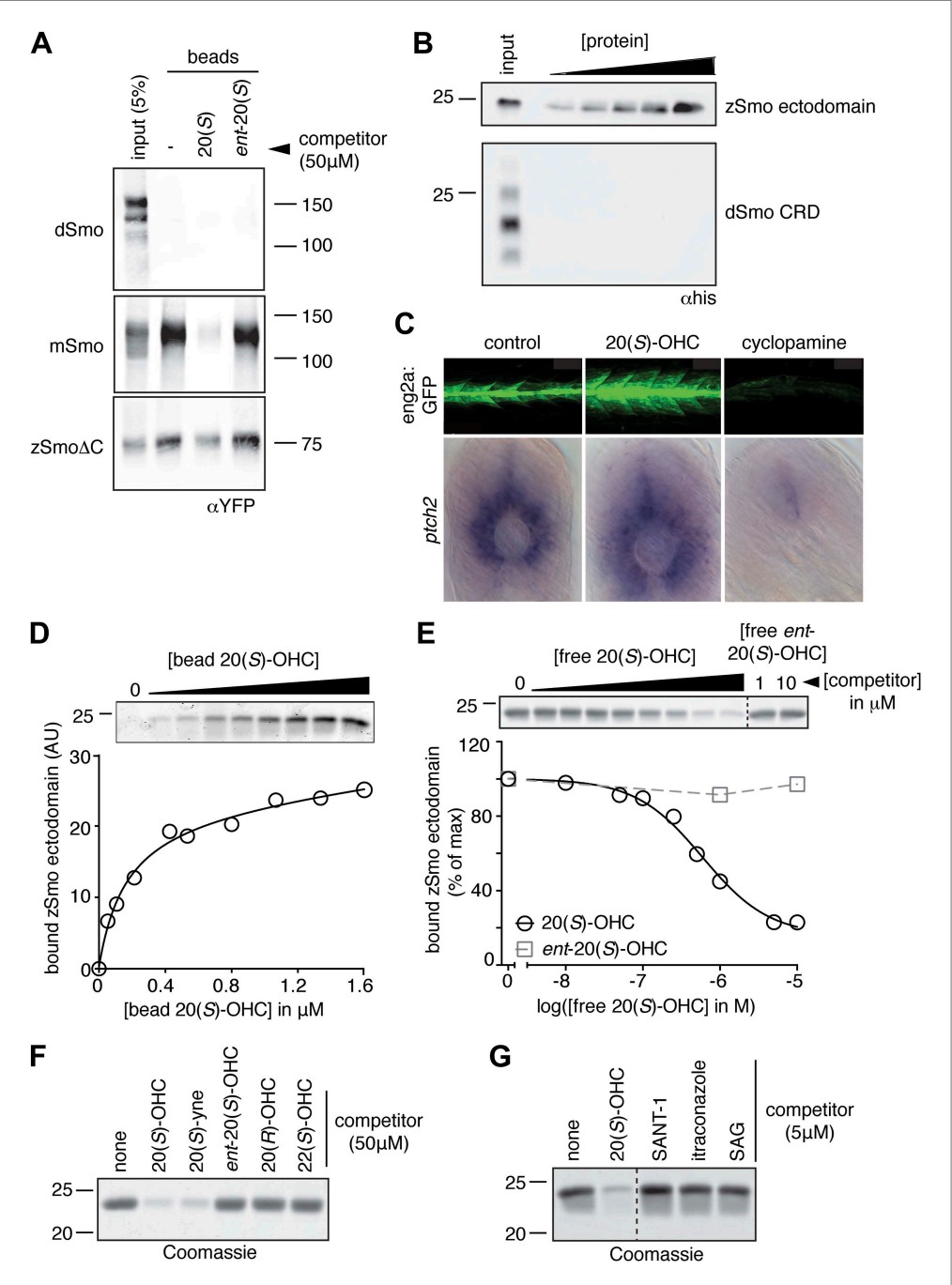

**Figure 4**. The Smo-oxysterol interaction is conserved in vertebrates. (**A**) The interaction of 20(*S*)-OHC beads with full-length mSmo, full-length Drosophila Smo (dSmo) or zebrafish Smo (zSmo) carrying a truncation of the intracellular C-terminal tail (zSmoΔC) was tested in the presence of free 20(*S*)-OHC or its enantiomer. (**B**) The zSmo ectodomain (which includes the CRD) can bind to 20(*S*)-OHC beads, but the dSmo CRD cannot. (**C**) Zebrafish embryos (30hpf) carrying a GFP transgene driven by the *engrailed2a* promoter were treated with 20(*S*)-OHC (50 μM) or cyclopamine (40 μM) and assessed for GFP expression by fluorescence and *ptch2* expression by in situ hybridization. See **Figure 4—figure supplement 1** for quantitation. (**D**) A binding curve ($K_d$ ~170 nM) for the zSmo ectodomain-20(*S*)-OHC interaction was measured by incubating a fixed amount of protein with increasing amounts of bead-immobilized sterol. The amount of zSmo ectodomain captured on the beads (shown in the graph) was quantitated from a coomassie-stained SDS-PAGE gel shown above. (**E**) Binding of the zSmo ectodomain to 20(*S*)-OHC beads was inhibited in a dose-responsive fashion by free 20(*S*)-OHC but not by its enantiomer. *Figure 4. Continued on next page*

*Figure 4. Continued*

(**F** and **G**) Coomassie-stained SDS-PAGE gels show the amount of zSmo ectodomain captured on 20(*S*)-OHC beads in the presence of various oxysterols (**F**) or Smo ligands (**G**).

The following figure supplements are available for figure 4:

**Figure supplement 1**. 20(*S*)-OHC activates Hedgehog signaling in zebrafish embryos.

---

(**Table 1**, **Figure 5—figure supplement 1A**). Refinement resulted in an R-factor of 21.6% (R-free: 26.0%) with two zSmo molecules in the crystallographic asymmetric unit, each composed of a well-defined model that included residues 41–158 (root mean square deviation [RMSD]: 0.60 Å for 118 Cα positions, **Figure 5—figure supplement 1B**). Although we set-up crystallization trials with the entire zSmo ectodomain (residues 29–212), the N- and C-terminal regions could not be traced due to missing electron density and thus were not included in the final model (**Figure 5—figure supplement 2**). The portion of the zSmo ectodomain spanning residues 41–158 (visible in our structure) shows sequence similarity to the previously identified CRD in the Fz protein family (**Dann et al., 2001**) and thus will hereafter be called the zSmo CRD.

The small interface between the two zSmo CRD molecules observed in the asymmetric unit of the crystal (buried surface area of 490 Å²) and a crystal contact formed by a zinc ion bonded to three

---

**Table 1.** Data collection and refinement statistics

|  | SeMet-substituted zSmo CRD | Native zSmo CRD |
|---|---|---|
| Data collection |  |  |
| Beamline | ESRF-ID23-EH1 | DIAMOND I03 |
| Wavelength | 0.979 | 1.000 |
| Space group | P4$_3$2$_1$2 | P4$_3$2$_1$2 |
| Cell Dimension (Å) | a, b = 68.2; c = 92.3 | a, b = 68.6; c = 95.3 |
| Resolution | 48.0–2.3 (2.36-2.30) | 31.0–2.6 (2.67-2.60) |
| Completeness (%) | 99.2 (92.8) | 98.9 (94.7) |
| Unique reflections | 10,085 | 7307 (491) |
| $R_{merge}$ (%) | 10.0 (92.0) | 13.4 (64.8) |
| I/σ(I) | 25.5 (3.9) | 14.4 (2.4) |
| Multiplicity | 26.5 (23.7) | 8.8 (6.0) |
| Refinement |  |  |
| Resolution range (Å) | 30.50–2.30 | 31.00–2.60 |
| No. reflections | 9576 | 7275 |
| $R_{work}$ (%) | 23.6 | 21.6 |
| $R_{free}$ (%) | 28.4 | 26.0 |
| No. atoms (protein/Zn/Na/water) | 1880/1/3/28 | 1880/1/2/48 |
| B-factors (Å²) (protein/Zn/Na/water) | 57/60/43/48 | 40/32/27/30 |
| r.m.s. deviations |  |  |
| Bond lengths (Å) | 0.012 | 0.004 |
| Bond angles (°) | 1.604 | 0.714 |
| Ramachandran statistics |  |  |
| Favored (%) | 96.5 | 97.9 |
| Disallowed (%) | 0.4 | 0 |

Each structure was determined from one crystal. Numbers in parentheses refer to the highest resolution shell. $R_{free}$ equals the *R*-factor against 5% of the data.

different protein chains (one chain A and two chain B molecules; *Figure 5—figure supplement 1C*) suggested that the dimeric arrangement observed in the crystal is not likely to be of functional significance. In agreement with this crystal packing analysis, purified zSmo ectodomain behaved as a monomer in solution at low concentration (5 µM) when assessed using multi angle light scattering (*Figure 5—figure supplement 1D*).

The zSmo CRD monomer adopts a globular fold composed of four α helices (α1: residues Q77-N92; α2: P94-Y108; α3: Q122-N130; α3′: S133-E138) and a short two-stranded β sheet (β1: K43-S45 and β2: K116-E118; *Figure 5A* and *Figure 5—figure Supplement 2*). This arrangement is stabilized by five disulfide bridges (labeled *, I, II, III, IV in *Figure 5A*). Disulfide bridges I, II, III, and IV lock the four helices together into a tight bundle, whereas disulfide bridge *, formed by the N- and C-terminal cysteines, orients the termini in close proximity and away from the helical bundle (*Figure 5A*). Structure-based evolutionary analysis of zSmo CRD revealed that the closest structural relatives are the CRDs of Frizzled 8 (Fz8; *Dann et al., 2001*; *Janda et al., 2012*), secreted Frizzled-related protein 3 (sFRP3, *Dann et al., 2001*) and muscle-specific kinase (MuSK, *Stiegler et al., 2009*), shown clustered in the blue branch in *Figure 5B* (*Figure 5—figure supplement 3*). These three structures show a similar helical bundle arrangement compared to the zSmo CRD, with the exception of a rearrangement of helix α3 and α3′, which forms a continuous helix in Fz8 and sFRP3. Strikingly, 4 out of 5 disulfide bridges are highly conserved (I, II, III, and IV), retaining the overall fold of the helix bundle. Only one disulfide bridge (labeled with an asterisk * in *Figure 5A*) is not conserved, resulting in a rearrangement of the relative orientations of the two termini compared to the zSmo CRD (*Figure 5C–E*).

Using structure fold recognition methods, Bazan and de Sauvage identified an additional group of Fz-like CRD containing proteins (*Bazan and de Sauvage, 2009*). These include the Niemann-Pick C1 protein (NPC1) and the riboflavin-binding protein (RFBP). Our evolutionary structural analysis confirmed their findings and allowed us to add the folate receptor α (FRα) to this group (*Figure 5B*, red branch and *Figure 5—figure supplement 3*) (*Chen et al., 2013*). Structural comparison of these proteins to the zSmo CRD revealed the common features identified for Fz-like CRDs, namely the helical bundle (formed by helices α1, α2 and α3) and the four conserved disulfide bonds that stabilize the fold and the relative orientations of the helices (*Figure 5F–H*).

## Mapping the Smo oxysterol binding site

A common feature of the Fz-like CRD family members is their ability to bind small, hydrophobic molecules in a pocket formed by the core helices α1, α2 and α3. While NPC1, RFBP and FRα bury their respective ligands in the protein core (cholesterol in NPC1, riboflavin in RFBP and folate in FRα) with the help of extensive protrusions from the core CRD fold (shown in gray in *Figure 5F–H*), Fz8, the closest structural homolog of the zSmo CRD structure, binds the palmitoleyl moiety covalently linked to Wnt proteins in a shallow groove (*Figure 5C*; *Janda et al., 2012*). To investigate the putative oxysterol binding site in the Smo CRD, we calculated the volumes of potential binding pockets in our zSmo CRD structure. The most prominent groove is indeed located at an equivalent position to the Fz8 palmitoleyl-binding groove (*Figure 6A,B*). The residues forming this groove are highly conserved in all vertebrate Smo CRDs (*Figure 6C* and *Figure 5—figure supplement 2*), and the volume (551 Å$^3$) and shape of the groove is sufficient for 20(*S*)-OHC binding. Computational docking using AutoDock (*Morris et al., 2009*) showed that the hydrophobic groove on the zSmo CRD surface (*Figure 6A*) can accommodate 20(*S*)-OHC with a favorable free energy of binding (*Figure 6—figure supplement 1A–C*). The four rings of the oxysterol are predicted to lie on the base of the groove lined with zSmo residues W87 and L90 and make additional potential hydrophobic interactions with zSmo residues M86, G89, Y108, G140, P142 and F144.

To test this model for the oxysterol-binding pocket, we mutated Smo residues that map to this pocket and, as controls, other residues that point away from the pocket or that are on the opposite face of the molecule (*Figure 6C,D*). All mutations were made in full-length mouse Smo, and mutant proteins were tested for binding to 20(*S*)-OHC beads after detergent-solubilization from membranes (*Figure 6E*). *Figure 6D* shows corresponding mouse and zebrafish residue numbers, and hereafter the residues are numbered according to the mouse sequence. Only those mutants that fractionated as a doublet on an SDS-PAGE gel were evaluated because this property demonstrates post-Golgi trafficking and hence correct folding (*Figure 1C*). Mutations in residues on the opposite face of the putative sterol binding pocket (E162A, P120A/E/G, P128S/E/R, P88N, L150A/D/S) or at the periphery of the pocket (R165A/E and N118A) did not disrupt binding to 20(*S*)-OHC beads (*Figure 6E* and *Figure 6—figure*

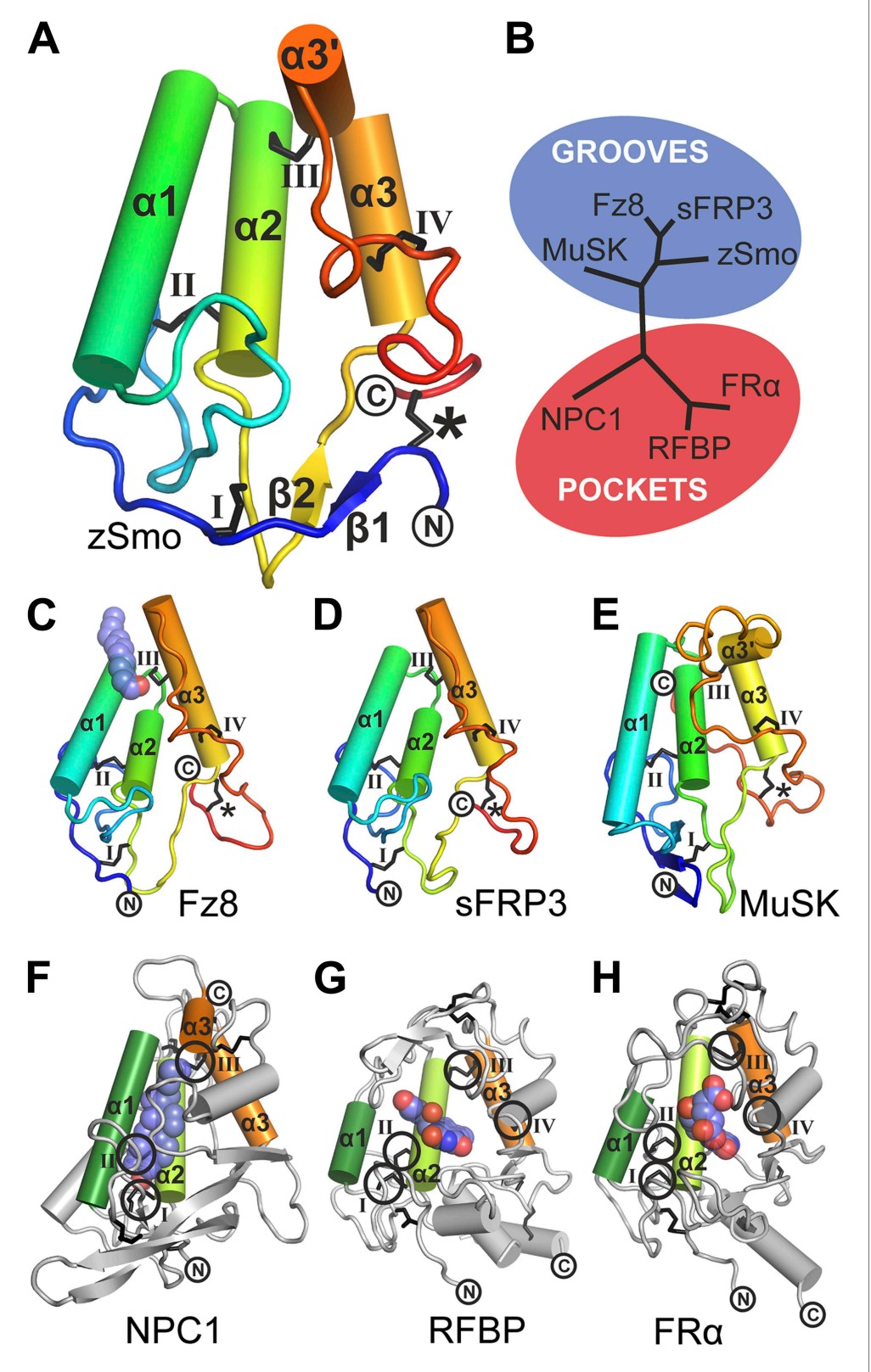

**Figure 5**. Structural analysis of the zebrafish Smo CRD. (**A**) Ribbon diagram of zSmo CRD in rainbow coloring from blue (N-terminus) to red (C-terminus) with the secondary structure elements numbered. The four disulfide bridges (black sticks) conserved in all Fz-like CRDs are depicted with Roman numerals, and the non-conserved disulfide

*Figure 5. Continued on next page*

*Figure 5. Continued*

bridge is marked with an asterisk (*). N- and C-termini are labeled. (**B**) Structural phylogenetic analysis of the CRDs. Structural superposition of CRDs from zSmo, Frizzled 8 (Fz8, PDB ID 4F0A, *Janda et al., 2012*), secreted Frizzled-related protein 3 (sFRP3, PDB ID 1IJX, *Dann et al., 2001*), muscle-specific kinase (MuSK, PDB ID 3HKL, *Stiegler et al., 2009*), Niemann-Pick C1 protein (NPC1, PDB ID 3GKI, *Kwon et al., 2009*), riboflavin-binding protein (RFBP, *Monaco, 1997*), and folate receptor α (FRα, PDB ID 4LRH, *Chen et al., 2013*) were superimposed using SHP (*Stuart et al., 1979*; *Riffel et al., 2002*). CRDs that form ligand-binding pockets (red background) or grooves (blue background) form two distinct evolutionary branches. In addition, CRDs show distant structural similarity to the extracellular domains of glypicans (*Pei and Grishin, 2012*). However, analysis of the crystal structures of glypicans Dally-like protein and glypican 1 revealed no apparent grooves or pockets that could accommodate small molecules (*Kim et al., 2011*; *Svensson et al., 2012*) and thus were not included in our structural analyses. (**C**–**H**) Ribbon diagrams of superimposed Fz-like CRD domains from the structural phylogenetic analysis in (**B**). (**C**) Fz8-palmitoleyl complex, (**D**) sFRP3, (**E**) MuSK, (**F**) NPC1-cholesterol complex, (**G**) RFBP-riboflavin complex, (**H**) FRα-folate complex. Color coding and labeling follows (**A**). Ligands are shown as spheres in atomic coloring (carbon: slate; oxygen: red; nitrogen: blue). In (**F**–**H**) the conserved disulfide bridges are highlighted with a circle. NPC1 (**F**) does not contain disufide bridge IV.

The following figure supplements are available for figure 5:

**Figure supplement 1**. Electron density of the zSmo CRD structure and oligomeric state of the zSmo ectodomain.

**Figure supplement 2**. Sequence alignment of the ectodomains of Smo family members and the CRD of mFz8.

**Figure supplement 3**. Structural comparison of CRDs.

*supplement 2*). In contrast, mutations in residues that frame the putative oxysterol pocket (L112A, L112D, G115F, L116A, Y134F, G166F, P168A, F170A) substantially reduced binding to 20(*S*)-OHC beads (*Figure 6E* and *Figure 6—figure supplement 2*). Taken together, our mutagenesis data support the structure-based model for the interaction between oxysterols and the Smo CRD.

To understand why *Drosophila* Smo does not bind oxysterols, we constructed a homology model of the dSmo CRD based on the zSmo structure (*Figure 6—figure supplement 1D*). Despite the notable sequence identity between zebrafish and *Drosophila* Smo CRDs (~42%) and the conserved disulfide bond pattern, the homology model revealed a substantially different oxysterol-binding groove on the dSmo CRD surface. 5 out of 8 residues that are essential for vertebrate Smo interactions with oxysterols (zSmo residues M86, W87, G89, Y108 and G140) are different in dSmo (corresponding dSmo residues D129, Y130, A132, F151 and F187; *Figure 6—figure supplement 1D*), potentially providing an explanation for why dSmo does not bind to oxysterols.

Finally, we tested a subset of these mSmo mutants for their ability to rescue Hh signaling in Smo[−/−] cells treated with Shh, SAG or 20(*S*)-OHC. The mutations that preserved 20(*S*)-OHC binding also preserved mSmo responsiveness to all three agonists (*Figure 6F* and *Figure 6—figure supplement 2B*). The most informative mutations were G115F, P168A and Y134F, the last a conservative change that substitutes a *Drosophila* residue (F) for the corresponding mouse residue (Y). All three mutants were responsive to SAG, showing that they were not disabled, but demonstrated substantially reduced 20(*S*)-OHC binding and responsiveness, with the mSmo Y134F being completely unresponsive (*Figure 6F*). Interestingly, Shh-responsiveness was unaffected in mSmo G115F but significantly reduced in both mSmo Y134F and P168A. Finally, there were a few mutants (e.g., L116A) that did not show strong binding to 20(*S*)-OHC beads in our in vitro assay but still modestly responded to 20(*S*)-OHC when introduced into Smo[−/−] cells. This discrepancy may be due to the fact that our signaling assay in intact cells is more sensitive than the binding assay with solubilized proteins, which is conducted in the presence of high detergent to maintain Smo solubility after extraction from membranes.

## Oxysterol-based inhibitors that target the Smo CRD

The current generation Smo inhibitors that have entered the clinic, including the FDA-approved drug Vismodegib, all engage the 7TM site on Smo (*Frank-Kamenetsky et al., 2002*). However, mutations that prevent drug binding or drug activity can lead to clinically relevant resistance to these agents (*Yauch et al., 2009*; *Dijkgraaf et al., 2011*). Antagonists that engage the oxysterol binding site in the CRD would represent an orthogonal strategy for Smo inhibition.

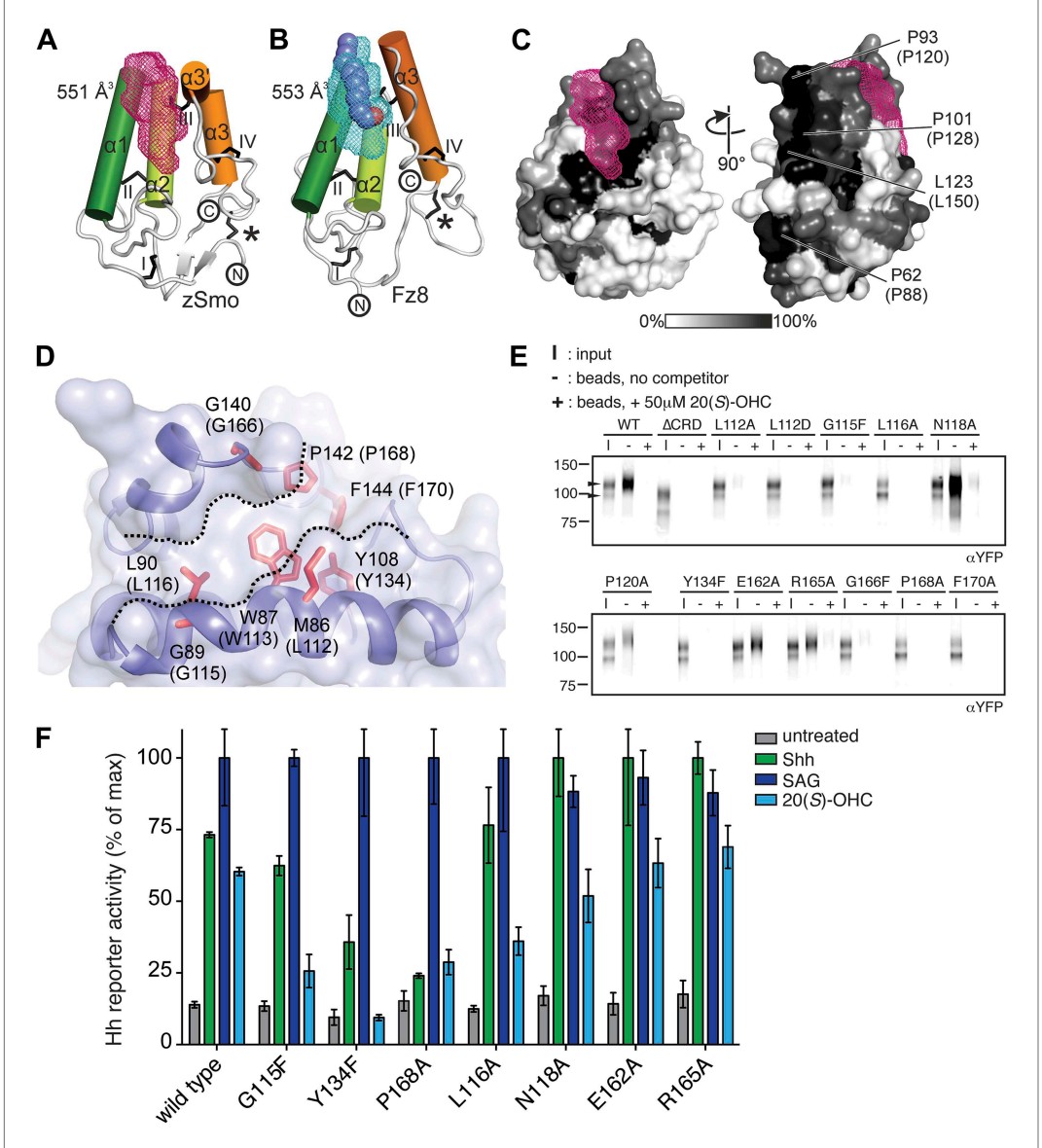

**Figure 6**. Mapping and analysis of the zSmo oxysterol binding site. (**A** and **B**) Ribbon representations of the zSmo CRD (**A**) and the Fz8 CRD-palmitoleyl (**B**) structures. View and presentation follows *Figure 5A*. The palmitoleyl-binding pocket of Fz8 is depicted as a cyan wire mesh and the corresponding pocket in the zSmo CRD structure is in red wire mesh. Volumes were calculated using the program Volumes (RE Esnouf, unpublished), with a 1.4 Å probe radius. The palmitoleyl moiety is shown as slate spheres. (**C**) The solvent accessible surface of the zSmo CRD is color-coded according to residue conservation (from non-conserved, white, to conserved, black) based on alignments containing amino acid sequences from >80 vertebrate Smo proteins. The right panel is rotated 90° around the y-axis relative to the left panel. Residues on the opposite face of the oxysterol-binding pocket that were subjected to mutagenesis are labeled. (**D**) Close-up view of the potential 20(*S*)-OHC binding site in the zSmo CRD structure. Residues predicted to make contacts with 20(*S*)-OHC are shown in stick representation and highlighted in red. Boundaries of the hydrophobic groove are marked with dotted lines. zSmo residues are numbered, with the corresponding mSmo residues in parentheses. (**E**) The indicated full-length mSmo point mutants were tested for their interaction with 20(*S*)-OHC beads in the absence or presence of free 20(*S*)-OHC competitor. Well-folded Smo mutants ran as a double band on a 4–12% Bis-Tris gradient gel (arrowheads), with only the slower-migrating species being captured on 20(*S*)-OHC beads. (**F**) A Hh reporter assay was used to measure signaling in Smo$^{-/-}$ cells transiently transfected with constructs encoding various mSmo point mutants and then treated (48 hr) with Shh, SAG (100 nM) or 20(*S*)-OHC (10 μM). The maximum reporter response for each mutant was set to 100%. Error bars denote S.D. (n = 3).

The following figure supplements are available for figure 6:

**Figure supplement 1**. Molecular modeling analysis of the zSmo CRD.

**Figure supplement 2**. Mutagenesis of the putative oxysterol binding site in the mSmo CRD.

To design such inhibitors, we considered two observations from our prior structure–activity relationship (SAR) studies on 20(S)-OHC (**Nachtergaele et al., 2012**). First, the stereochemistry at position 20 that determines the spatial relationship between the ring system and the *iso*-octyl chain is critical for the ability of 20(S)-OHC to activate Smo, since 20(R)-OHC is inactive. Second, the replacement of the *iso*-butyl group at the end of the *iso*-octyl chain with an alkyne group increased Hh-activation potency by ~10-fold (**Figure 7A**). Starting from this high-potency Smo activator 20(S)-yne, we inverted the stereochemistry at position 20 to make 20(R)-yne or oxidized the hydroxyl group to a ketone, changing carbon 20 to a planar $sp^2$ hybridized center, to make 20-keto-yne (**Figure 7A**). Both molecules blocked the binding of mSmo CRD-Fc to 20(S)-OHC beads but did not affect the binding of a fluorescent

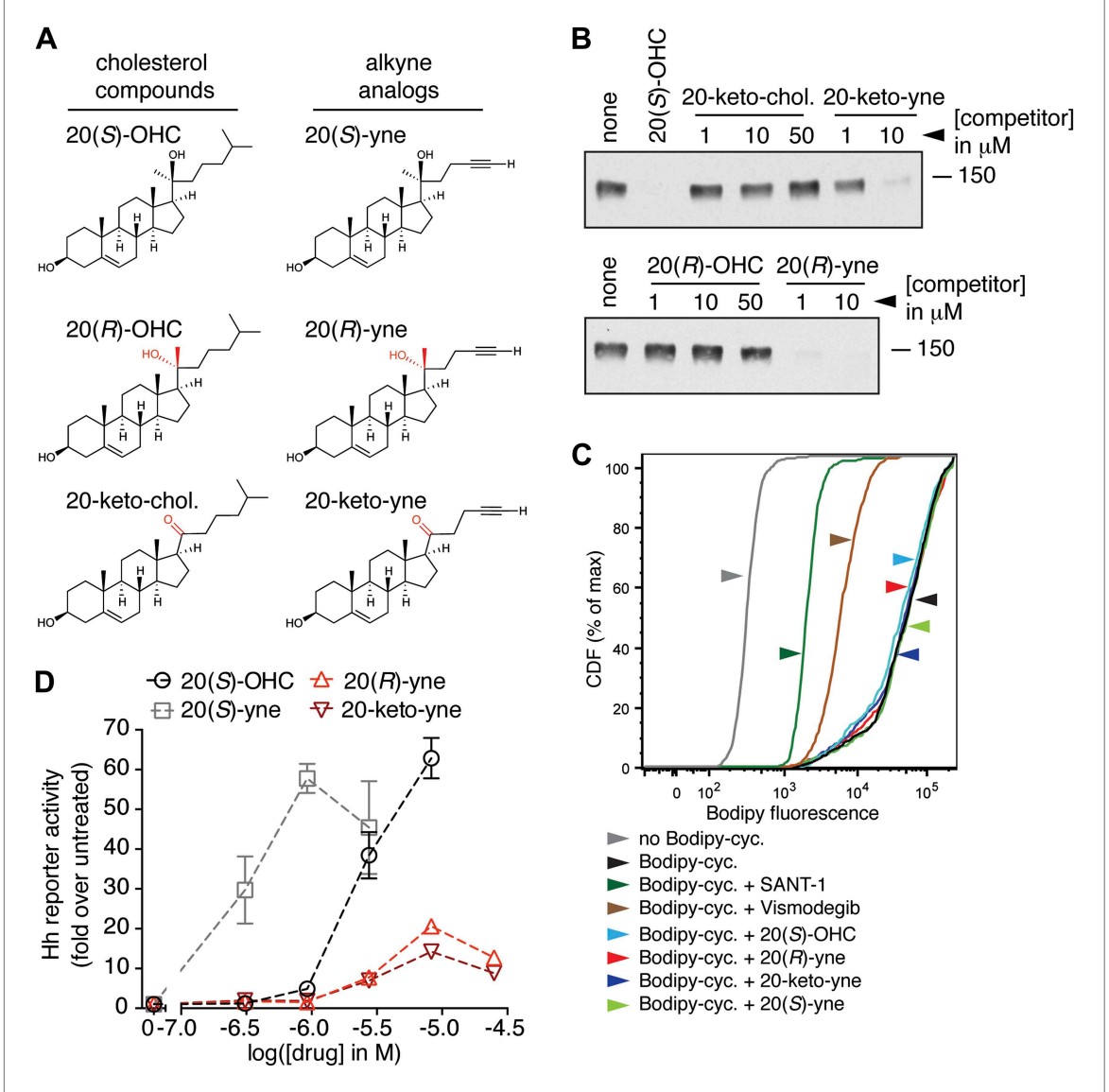

**Figure 7**. Partial agonists that target the Smo CRD. (**A**) Structure and synthetic logic for 20(R)-yne and 20-keto-yne. 20(R)-OHC and 20-keto-cholesterol are related analogs that lack the alkyne moiety. (**B**) Immunoblots show the amount of mSmo CRD-Fc captured on 20(S)-OHC beads in the presence of the indicated oxysterols added as competitors. (**C**) Binding of bodipy-cyclopamine to cells expressing full-length mSmo was determined by FACS in the presence of various Smo ligands. The bodipy-cyclopamine fluorescence in a cell population is expressed as a cumulative distribution function (CDF), which denotes the percentage of cells that show a given level of fluorescence or lower. Bodipy-cyclopamine binding can be competed by SANT-1 and Vismodegib, two 7TM site ligands, but not by any of the CRD-binding oxysterols. (**D**) Hh reporter activity in cells treated with increasing concentrations of the indicated oxysterols.

cyclopamine derivative (bodipy-cyclopamine) to Smo-expressing cells, showing that they engaged the CRD site but not the 7TM site (*Figure 7B,C*). The alkyne group was an important structural feature required for competition, as both 20(*R*)-OHC and 20-keto-cholesterol (*Figure 7A*) failed to inhibit the CRD–20(*S*)-OHC interaction (*Figure 7B*).

Despite binding to the mSmo CRD, 20(*R*)-yne and 20-keto-yne were weak activators of signaling in the absence of Shh, reinforcing the importance of stereochemistry at position 20 for Smo activation (*Figure 7D*). However, both molecules inhibited signaling induced by the native ligand Shh, the CRD agonist 20(*S*)-OHC or the 7TM agonist SAG (*Figure 8A–C* and *Figure 8—figure supplement 1*). Both the molecules also reduced signaling by mSmoM2, a constitutively active, oncogenic Smo mutant (*Taipale et al., 2000*), and mSmo D477H, the mouse version of a human Smo mutant that is resistant to the FDA-approved drug Vismodegib (*Figure 8D,E*) (*Yauch et al., 2009*). We hereafter call these molecules oxysterol-based inhibitors or OBIs. This activity profile shows that the OBIs are CRD-targeted partial agonists of Smo that can reduce signaling by Smo activators and by clinically relevant Smo mutants. Our OBIs seem to inhibit Smo by a different mechanism compared to another recently reported CRD antagonist, 22-azacholesterol, which does not block signaling induced by SAG or by mSmoM2 (*Nedelcu et al., 2013*). The broader Hh inhibitory activity of OBIs is instead reminiscent of the glucocorticoids Budesonide and Ciclesonide, which also fail to compete with cyclopamine for binding to Smo (*Wang et al., 2012*).

An early step in signaling that precedes transcription is the Shh-induced accumulation of Smo in the primary cilium (*Corbit et al., 2005*). Antagonists that bind to the 7TM site display striking differences in their impact on this key trafficking step. Cyclopamine and cyclopamine derivatives (that contain a sterol-like tetracyclic ring structure) do not block Smo ciliary accumulation (*Figure 8F*) and in fact can drive Smo accumulation in cilia even in the absence of Shh (*Rohatgi et al., 2009*). On the other hand, non-sterol 7TM antagonists like SANT-1 and Vismodegib prevent Shh-induced Smo accumulation in cilia (*Rohatgi et al., 2009*). The CRD-targeted OBIs both behaved like cyclopamine in this assay—they induced Smo accumulation in cilia when added alone and also did not block Shh-induced ciliary accumulation of Smo (*Figure 8F*). This similarity between the OBIs and cyclopamine led us to consider the possibility that cyclopamine might not be a pure 7TM inhibitor like SANT-1 and Vismodegib but instead may also engage the CRD. Indeed, unlike the non-sterol 7TM inhibitors (*Figure 2E*), cyclopamine blocked the interaction between the mSmo CRD-Fc and 20(*S*)-OHC beads (*Figure 8G*), suggesting that it is capable of binding the CRD in this in vitro assay.

## Discussion

Our work provides both structural and mechanistic insights into the enigmatic CRD of Smo in Hh signaling. The CRD of Fz proteins binds to Wnt ligands. While the Fz CRD is related to the Smo CRD, no protein ligand has been identified to date that directly binds to the Smo CRD, and its role in Smo function has not been defined. In *Drosophila*, deletion of the Smo ectodomain or the mutation of specific cysteine residues in the CRD completely inactivates the protein (*Nakano et al., 2004*). In contrast, in cultured mouse cells, ΔCRD-Smo has basal activity in overexpression experiments (*Murone et al., 1999*; *Taipale et al., 2002*). In zebrafish embryos, ΔCRD-Smo can rescue phenotypes dependent on low-level signaling but not on high-level signaling, and it shows higher levels of basal accumulation in cilia (*Aanstad et al., 2009*). Our work now shows that the Smo CRD in vertebrates binds to oxysterols and mediates the ability of these lipids to activate Hh signaling. Structure-guided mutagenesis studies revealed that the Smo CRD binds to 20(*S*)-OHC in the region that was previously identified as the binding site for small hydrophobic molecules in other CRDs, formed by the evolutionary conserved helical bundle of the CRD core. This supports the hypothesis that CRDs evolved from an ancestral domain that sensed hydrophobic molecules (*Bazan and de Sauvage, 2009*). Our structural analysis showed that the Smo CRD oxysterol binding site is most similar to the palmitoleyl-binding site in Fz CRDs (*Janda et al., 2012*); however, the binding grooves are built of divergent residues (*Figure 5—figure supplement 2* and *Figure 6—figure supplement 1*), suggesting that they accommodate different classes of hydrophobic ligands.

Smo activity can be regulated by two distinct binding sites in the CRD and the 7TM segments. Oxysterols and their derivatives regulate Smo through the CRD site, while SANT-1, SAG, and Vismodegib bind to the 7TM segment. Binding of agonists like 20(*S*)-OHC and 20(*S*)-yne to the CRD must be communicated to the 7TM helix bundle for transduction across the membrane. Indeed, while the 7TM

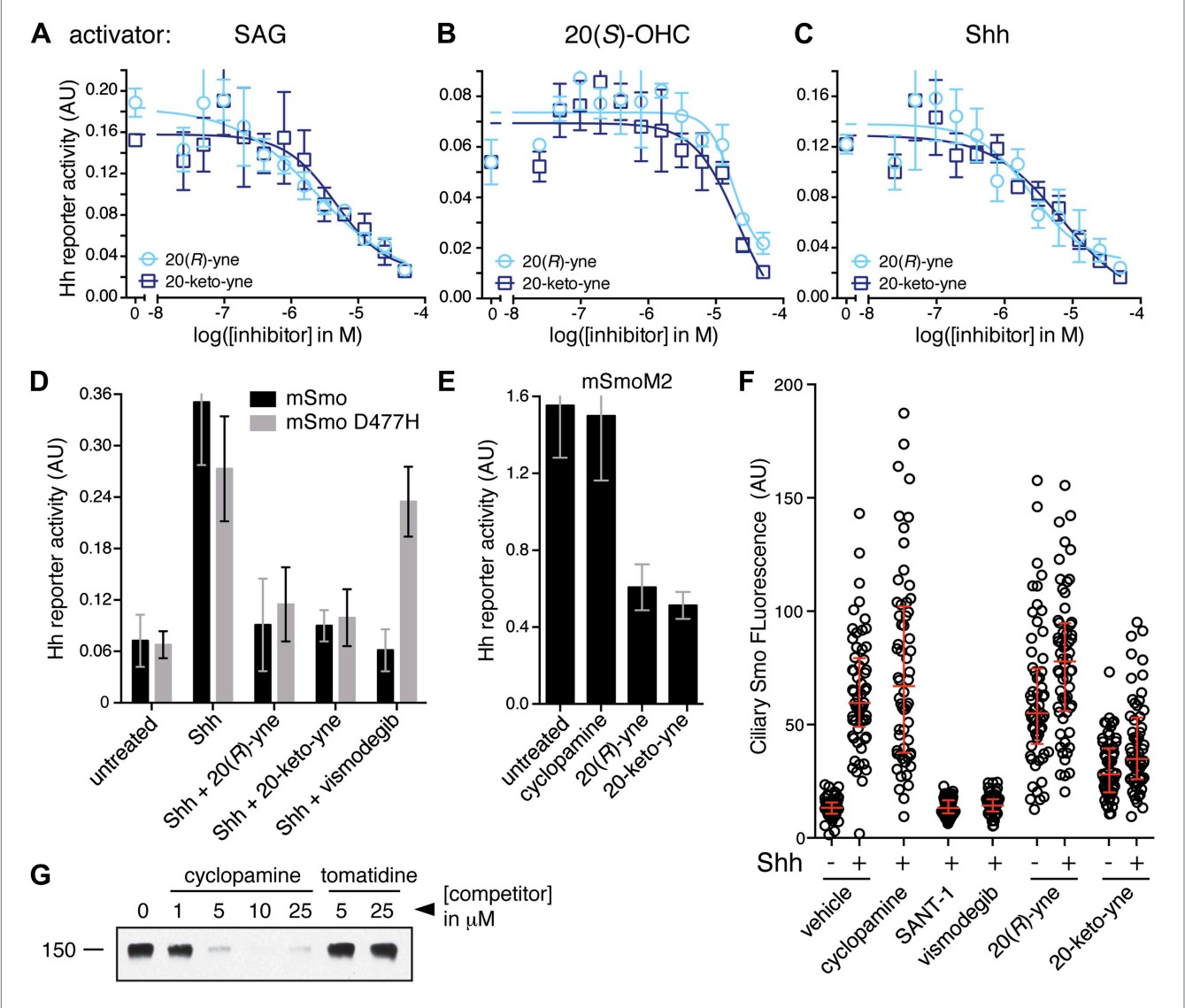

**Figure 8**. 20(R)-yne and 20-keto-yne can inhibit Hh signaling. (**A–C**) Hh reporter activation by 100 nM SAG (**A**), 5 μM 20(S)-OHC (**B**) and Shh (**C**) can be inhibited by both 20(R)-yne and 20-keto-yne. (**D**) Hh reporter activity in Smo^−/− cells transfected with wild-type mSmo or mSmo D477H and then treated (48 hr) with Shh in the presence of 20(R)-yne, 20-keto-yne (both at 25 μM) or vismodegib (100 nM). (**E**) Hh reporter activity in NIH 3T3 cells transfected with constitutively active, oncogenic mSmoM2 and then treated (12 hr) with 20(R)-yne and 20-keto-yne (10 μM each) or cyclopamine (1 μM). (**F**) Accumulation of endogenous mSmo in cilia of NIH 3T3 cells treated (4 hr) with the indicated Smo ligands in the presence or absence of Shh. Each point represents the Smo fluorescence in a single cilium and the red lines denote the median and the interquartile range of mSmo fluorescence (n = 60 for each condition). (**G**) The binding of mSmo CRD-Fc to 20(S)-OHC beads can be inhibited by cyclopamine but not by the structurally-related alkaloid tomatidine. Error bars denote S.D. (n = 3).

The following figure supplements are available for figure 8:

**Figure supplement 1**. Table of IC50 values for the OBIs.

and CRD sites are separable, the dramatic synergy between 20(S)-OHC and SAG we have previously reported suggests a positive allosteric link between these domains (**Nachtergaele et al., 2012**). This synergy also implies that 7TM and CRD ligands can bind to Smo simultaneously.

A speculative but intriguing insight into the interaction between the CRD and 7TM domains comes from our unexpected finding that the Hh inhibitor cyclopamine, an established 7TM ligand, also inhibits the binding of the isolated CRD to 20(S)-OHC beads. This result was unexpected because oxysterols

do not decrease the binding of bodipy-cyclopamine to cells expressing full-length Smo (*Figure 7C* and *Dwyer et al., 2007*). We believe that this discrepancy is due to the fact that the cell-based bodipy-cyclopamine binding assay is mostly measuring the interaction between cyclopamine and its high affinity ($K_d$~10 nM, *Rominger et al., 2009*) binding site in the 7TM domain. Cell (or membrane) binding assays can often miss lower affinity (~1–10 µM) interactions, which can be detected by ligand affinity chromatography assays (*Phizicky and Fields, 1995*). It is also possible that cyclopamine binds the CRD more weakly when the CRD is embedded in the context of the whole protein.

As noted above, cyclopamine is a sterol that induces the accumulation of Smo in primary cilia, both properties that distinguish it from the pure 7TM site inhibitors SANT-1 and Vismodegib. One possibility is that cyclopamine can bridge the two ligand-binding sites on Smo and engage both a high-affinity interaction with the 7TM segment and a lower affinity interaction with the CRD. Alternatively, two molecules of cyclopamine could engage the CRD and 7TM sites separately or cyclopamine could be involved in a 'hand-off' interaction between the CRD and the 7TM segments analogous to the manner in which cholesterol is transferred between NPC2 and NPC1 (*Kwon et al., 2009*; *Wang et al., 2010*). While the relevance of this interaction for the inhibition of Smo by cyclopamine in cells remains to be established, the puzzling ability of cyclopamine to induce Smo accumulation in cilia (while inhibiting Smo activity) may be related to its ability to engage the CRD. This represents a third mechanism by which ligands can engage Smo, one that is distinct from pure 7TM and CRD ligands. Interestingly, glucocorticoids have been shown to fall into two distinct classes of Smo modulators—cyclopamine-competitive ligands that presumably bind to the 7TM potentiate signaling and a second class of inhibitors that do not compete with cyclopamine but appear to engage a distinct site (*Wang et al., 2012*).

The CRD of Smo is also important for signaling by Shh, since ΔCRD-Smo cannot be efficiently activated by either Shh or 20(*S*)-OHC but remains responsive to SAG. While we observed very little activation of ΔCRD-YFP-Smo by Shh (*Figure 3B*), another study (*Nedelcu et al., 2013*) reported that a ΔCRD-Smo-mCherry protein retained a low level of Shh responsiveness, suggesting that the CRD is not absolutely required for signaling initiated by Shh. This difference in the degree of Shh-responsiveness may be due to the position of the fluorescent protein tag, differences in the tendency of the YFP and mCherry tags to oligomerize or differences in the expression systems used in the two studies.

The striking decrease in Shh-responsiveness when the CRD is deleted raises two questions—does Ptch1 regulate Smo through the oxysterol binding site in the CRD and is 20(*S*)-OHC an endogenous ligand for Smo? Our mutagenesis of the putative oxysterol binding site in the CRD sheds light on the first question. We find mutations in the mSmo CRD (Y134F and G115F, *Figure 6F*) that can dissociate the Shh and oxysterol responses. These mutants fail to bind or respond to 20(*S*)-OHC but can still respond to Shh. The simplest interpretation of these data is that the endogenous Smo ligand regulated by Ptch1 does not bind Smo in precisely the same site as 20(*S*)-OHC. In fact, we have previously reported (*Nachtergaele et al., 2012*) that cyclopamine is much less potent against Shh-activated Smo compared to 20(*S*)-OHC-activated Smo, likely because the conformation adopted by Smo is different in response to these two agonists. Both of these findings suggest that 20(*S*)-OHC is not the Ptch1-regulated ligand that modulates Smo activity in response to Shh reception. It remains possible that a Shh-regulated ligand binds to the CRD in a manner that is distinct from that of 20(*S*)-OHC.

The CRD is required for Smo to adopt a fully active conformation in response to Shh (but it is dispensable when Smo is activated by the synthetic 7TM ligand SAG). In this view, the CRD would serve as a domain that allosterically activates the 7TM helix bundle in response to Shh. Some mutations (Smo Y134F, P168A) that abolish 20(*S*)-OHC responses do indeed substantially dampen the ability of Shh to activate Smo. The observation that CRD point mutations in Smo that block oxysterol binding also impair signaling by Hh ligands has been used to infer that oxysterol binding is required for physiological Smo signaling (*Nedelcu et al., 2013*). While this hypothesis has substantial implications for Hh regulation in development and cancer, it remains to be determined if the CRD site in cells is occupied by oxysterols or by a different ligand, or if perturbations in endogenous oxysterol levels can modulate Hh signaling. Finally, testing the activity of oxysterol binding site mutants in the context of embryonic development or Hh-driven tumors is essential for elucidating the physiological function of this site and whether it plays a role in graded, low-level or high-level signaling.

We have developed partial agonists of Smo that bind to the CRD. Understanding the structural and mechanistic basis for this partial agonism is an important future goal. Remarkably, the simple inversion

of the stereochemistry at C-20 converts a potent agonist into a weak partial agonist and an effective inhibitor of signaling. This stereochemical inversion presumably allows the molecule to trap Smo in a poorly active confirmation, likely one similar to that stabilized by cyclopamine, in which Smo is localized in cilia but is inactive. The structures of the OBIs suggest that Smo activation potential depends critically on the spatial orientation between the ring system and the *iso*-octyl chain of 20(*S*)-OHC. Regardless of the mechanism, inhibitors targeting the Smo CRD would provide an orthogonal approach to modulate Hh signaling in regeneration and cancer. Partial agonists offer the possibility of blocking unrestrained signaling (such as that seen in cancer) while preserving lower-level, physiological signaling (*Riese, 2011*). This ability to attenuate Smo activity may be useful since currently used Smo antagonists cause significant side-effects, leading nearly half of the patients in some trials to discontinue treatment (*Tang et al., 2012*).

Perhaps the most important question moving forward is to identify the Shh-regulated ligand that mediates the communication between Ptch1 and Smo and to understand how it regulates Smo through the 7TM and CRD sites. Structural studies of a Smo construct carrying both the 7TM segment and the CRD in complex with various ligands that engage either site or both sites will be essential to understand how Smo transmits the Hh signal across the membrane.

## Materials and methods

### Cells and reagents

NIH 3T3 and 293T cells were obtained from ATCC (Manassas, VA), and 293F cells were obtained from Life Technologies (Grand Island, NY). The production of Smo$^{-/-}$:YFP-mSmo, Smo$^{-/-}$:ΔCRD-YFP-mSmo and Smo$^{-/-}$:ΔC-YFP-mSmo stable lines is described below. SAG ($\geq$95%) was from Enzo Life Sciences (Farmingdale, NY), cyclopamine ($\geq$98%) from Toronto Research Chemicals (Toronto, Ontario, Canada), Itraconazole ($\geq$98%) from Sigma (St. Louis, MO) and SANT-1 ($\geq$95%) from EMD Millipore (Billerica, MA). All sterols except *ent*-20(*S*)-OHC, 20(*S*)-yne, 20(*R*)-OHC, 20(*R*)-yne, 20-keto-cholesterol, 20-keto-yne and 20(*S*)-amine were purchased from Steraloids (purity $\geq$98%) (Newport, RI).

### Constructs

All mSmo mutants were made using Quickchange or PCR methods in the context of a previously described construct encoding full-length mouse Smo (UniProt P56726) with the coding sequence for yellow fluorescent protein (YFP) inserted immediately after the signal sequence (pCS2:YFP-mSmo; *Rohatgi et al., 2009*). The ΔCRD-YFP-mSmo construct lacked amino acids (a.a.) 68–184, and the ΔC-YFP-mSmo construct was truncated after a.a. 574. The construct encoding mSmo CRD-Fc was made by cloning the mSmo sequence encoding the CRD (a.a. 1–183) into a pCX vector carrying a C-terminal human Fc tag. Constructs for mammalian expression of the extracellular region of zebrafish Smo (UniProt Q90X26; zSmo ectodomain: a.a. 29–195), of a C-terminal, intracellular truncation of zSmo (UniProt Q90X26; zSmo ΔC: a.a. 29-624) or the CRD of *Drosophila* Smo (UniProt P91682; dSmo CRD: a.a. 32–204), fused C-terminally with either a hexa-histidine, mono Venus or 1D4 epitope-tag that can bind selectively the Rho 1D4 antibody (*Molday and MacKenzie, 1983*), were cloned into the pHLsec vector (*Aricescu et al., 2006*). A construct for bacterial expression of the extracellular region of zebrafish Smo (UniProt Q90X26; zSmo-ectodomain: a.a. 29–212), fused C-terminally with a with a hexa-histidine (His6) tag, was cloned into the pET22b vector.

### Stable cell lines

Stable cell lines expressing YFP-mSmo, ΔCRD-YFP-mSmo and ΔC-YFP-mSmo were made by infecting Smo$^{-/-}$ cells with a retrovirus carrying these constructs cloned into pMSCVpuro. Retrovirus was generated by transfecting the MSCV:YFP-mSmo constructs into Bosc23 cells. The virus-containing media were used to infect Smo$^{-/-}$ MEFs, and stable integrants were selected with puromycin and cloned by FACS.

### Chemical synthesis (general methods)

We have previously reported the chemical synthesis of *ent*-20(*S*)-OHC, 20(*S*)-yne, 20(*R*)-OHC and 20-keto-cholesterol (*Nachtergaele et al., 2012*). Full synthetic procedures are provided below for **20-keto-yne**, **20(*R*)-yne**, and **20(*S*)-amine**. Melting points were determined on a Kofler micro hot stage and were uncorrected. NMR spectra were recorded in CDCl$_3$, at 300 MHz ($^1$H) or 75 MHz ($^{13}$C). Chemical shifts (δ) were reported downfield from internal Me4Si (δ: 0.00). HR FAB-MS determinations were made with the use of JEOL MStation (JMS-700) Mass Spectrometer, matrix m-nitrobenzyl alcohol,

with NaI as necessary, using mass spectrometry facilities located at the University of Missouri–St. Louis. HIRES-MS determinations were made with the use of Thermo Orbitrap Velos Mass Spectrometer, using the facilities located at Washington University in St. Louis. IR spectra were recorded as films on a NaCl plate or in KBr. Elemental analyses were carried out by M–H–W laboratories. Optical rotations were measured on a Perkin-Elmer polarimeter, Model 341. Chromatography was performed using flash chromatography grade silica gel (32–63 µm; Scientific Adsorbents, Atlanta, GA). Dichloromethane was distilled over CaH prior to application. Tetrahydrofuran was distilled over Na/benzophenone just prior to application. All other chemicals were used as purchased without further purification. Organic extracts were dried over anhydrous $Na_2SO_4$.

## 20(*S*)-OHC bead synthesis

20(*S*)-amine was prepared as a 10 mM stock in 1:1 chloroform/methanol. For each coupling reaction, 250 µl (packed volume) of FastFlow 4 NHS-activated sepharose (GE Healthcare, San Francisco, CA) was washed extensively into DMSO. 300 µl of DMSO, 2.5 µl of the 10 mM 20(*S*)-amine stock and 1.5 µl of triethylamine were added to the washed resin, and the reaction was rotated for 4 hr at room temperature, protected from light. After coupling, the beads were spun down, the supernatant removed and 1 ml of 5% ethanolamine in DMSO was added to block the remaining free reactive sites (4 hr, room temperature, protected from light).

## Hedgehog reporter assays

For reporter assays in NIH 3T3 cells, a 10-cm plate of cells was transfected with 8 µg of a 4:1 wt/wt ratio of firefly luciferase reporter driven by an 8xGli-responsive promoter (*Sasaki et al., 1997*) and a Renilla luciferase reporter driven by a constitutive TK promoter (Promega, Madison, WI). The next day, transfected cells were seeded into a 96-well plate, grown to confluence, and treated overnight with drugs diluted in media containing 0.5% fetal bovine serum (FBS). For reporter assays in Smo$^{-/-}$ cells, 25,000 cells per well were seeded in a 24-well plate 24 hr prior to transfection. The next day, after a media replacement step, each well was transfected with 1 ng Smo construct and 500 ng of the reporter mix described above, using Xtreme Gene HP (Roche, Mannheim, Germany). After overnight transfection, the media were once again changed to fresh media. Cells were grown to confluence and treated with drugs diluted in media with 0.5% FBS for 48 hr. Activity of both reporters was measured using the Dual-Luciferase Reporter kit (Promega) and read on a Synergy H1 Hybrid Multi-Mode Microplate Reader (BioTek, Winooski, VT). The Gli luciferase to Renilla luciferase ratio is reported as 'Hedgehog reporter activity'. Each experiment, which included three technical replicates, was repeated at least three times.

## Protein expression and purification of mSmo CRD

pCX-mSmo CRD-Fc was produced by secretion (96 hr) into the media of 293F suspension cells (Life Technologies, Grand Island, NY) transfected with an expression construct. The collected media were cleared by centrifugation (10 min, 1000×*g*, 4°C), adjusted to pH 8.5, filtered through a 0.22 µm PVDF membrane and applied to a 1 ml Protein A Hitrap column (GE Healthcare). mSmo CRD-Fc was eluted from the Protein A column with 100 mM citrate pH 3.5, immediately adjusted to pH 8.5 and then loaded on a Superose 6 (10/300, GE Healthcare) gel filtration column equilibrated in 20 mM Tris pH 8.5, 150 mM NaCl. Monodisperse protein that eluted as a sharp peak (*Figure 2A*) was collected and used for binding assays. The purified mSmo CRD could not be cleaved away from the Fc tag efficiently and thus was used in assays as the fusion. In addition, it could not be heated above 37°C prior to SDS-PAGE electrophoresis because it underwent irreversible aggregation.

## Expression and purification of the zSMO ectodomain and dSmo CRD from mammalian cells

The zSmo ectodomain and dSmo CRD were expressed by transient transfection in HEK-293T cells (using an automated procedure, *Zhao et al., 2011*). 5 days post-transfection, the conditioned medium was dialyzed (for 48 hr at 4°C), and the ectodomain constructs of zSmo or dSmo were purified by either immobilized Rho 1D4 antibody affinity chromatography using CNBr-Activated Sepharose (GE Healthcare) as described previously (*Molday and MacKenzie, 1983*) or IMAC using Talon beads (Clontech, Mountain View, CA). Proteins were concentrated and further purified by size-exclusion chromatography (Superdex 200 16/60 column; GE Healthcare) in buffer containing 10 mM HEPES, pH 7.5, 150 mM NaCl.

## Expression and purification of native and selenomethionine (SeMet)-substituted zSmo ectodomain from *E. coli*

The zSmo ectodomain used for crystallization and oxysterol binding assays was expressed in *E. coli* Rosetta(DE3)pLysS cells (Novagen/EMD Millipore) as inclusion bodies and purified as follows (protocol adapted from *Brown et al. (2002)*). After cell lysis, the inclusion body pellets were washed four times and then solubilized in 8 M urea, 50 mM Tris-HCl, pH 8, and 100 mM NaCl. The solubilized protein was then purified via IMAC (Ni-Sepharose FastFlow; GE Healthcare) under denaturing conditions. After IMAC purification the eluted protein was reduced with 10 mM DTT and added drop-wise to 1 l of rapidly-stirring refold buffer (3 M urea, 150 mM Tris pH 8.5, 200 mM *L*-arginine, 1.5 mM reduced glutathione [GSH], 0.15 mM oxidized glutathione [GSSG]), which was then further stirred gently overnight at room temperature. The solution was then dialysed into 25 mM Tris pH 8.5, 10 mM NaCl at 4°C, filtered, loaded onto a 5 ml HiTrap QFF column (GE Healthcare) and eluted with an NaCl gradient (from 10 mM to 1 M NaCl). The eluted protein was concentrated and further purified via size exclusion chromatography (Superdex 75 16/60 [GE Healthcare] in 10 mM HEPES pH 7.5, 150 mM NaCl).

SeMet-labeled zSmo ectodomain was produced in *E. coli* strain B834 (DE3) (Novagen/EMD Millipore). Cells were grown in 2 l cultures at 310 K for 4 hr and after induction with 300 μM isopropyl β-D-1-thiogalactopyranoside, the temperature was then lowered to 298 K. Following incubation for further 20 hr, the cells were harvested and the protein was purified as described for the unlabeled zSmo ectodomain.

## Immunoblotting

Cultured cells stably expressing YFP-mSmo, ΔCRD-YFP-mSmo, or ΔC-YFP-mSmo were scraped into ice-cold PBS containing SigmaFast Protease inhibitor cocktail (Sigma) and collected as a pellet by centrifugation (1000×*g*, 10 min, 4°C). Cells were lysed (1 hr, 4°C) by agitation in modified RIPA buffer (50 mM sodium-Tris pH 7.4, 150 mM sodium chloride, 2% NP-40, 0.5% deoxycholate, 0.1% sodium-dodecyl sulfate [SDS], 1 mM dithiothreitol and the SigmaFast Protease inhibitor cocktail). After clarification (20,000×*g*, 45 min, 4°C), the protein concentration of each lysate was measured using the bicinchoninic acid assay (BCA, Pierce/Thermo Scientific, Rockford, IL). Lysate aliquots containing equal amount of total protein were fractionated on an 8% SDS-PAGE gel and transferred to a nitrocellulose membrane for immunoblotting with anti-Gli1 antibody (#L42B10, 1:500; Cell Signaling, Denvers, MA), anti-GFP antibody (1:5000; Novus, Littleton, CO), anti-Gli3 antibody (AF3690 1:200; R&D, Minneapolis, MN) and anti-p38 antibody (ab31828, 1:2000; Abcam, Cambridge, MA). In *Figures 1–4*, vertical dashed black lines represent non-contiguous lanes from the same immunoblot juxtaposed for clarity.

## Ligand affinity chromatography

293T cells transfected with constructs encoding mSmo variants were lysed in hypotonic SEAT buffer (250 mM sucrose, 1 mM EDTA, 10 mM acetic acid, 10 mM triethanolamine and the SigmaFast EDTA-Free protease inhibitor cocktail). After the removal of nuclei by centrifugation (900×*g*, 5 min, 4°C), membranes were pelleted by ultracentrifugation (95,000×*g*, 30 min) and solubilized in a n-dodecyl-β-D-maltopyranoside (DDM) extraction buffer (50 mM Tris pH 7.4, 150 mM NaCl, 10% vol/vol glycerol, 0.5% wt/vol DDM and the SigmaFast EDTA-Free protease inhibitor cocktail) for 2 hr at 4°C, followed by removal of insoluble material by ultracentrifugation (100,000×*g*, 30 min). This DDM membrane extract was incubated with 20(*S*)-OHC beads for 12 hr at 4°C to allow binding to equilibrium. After extensive washing, proteins captured on the beads were eluted with reducing SDS sample buffer. The presence of YFP-mSmo in these eluates was determined by quantitative immunoblotting with an anti-YFP antibody (NB600-308, 1:5000; Novus) and infrared imaging (Li-Cor Odyssey).

For ligand affinity chromatography with purified mSmo CRD-Fc or zSmo ectodomain, protein was diluted in 20 mM Tris pH 8.5, 150 mM NaCl, 0.3% octyl-glucoside prior to addition of competitors and 20(*S*)-OHC beads. After binding was allowed to proceed for 12 hr at 4°C, the resin was washed and captured protein was eluted as described above. The presence of mSmo CRD-Fc was measured by an anti-human HRP-coupled antibody (1:20,000) or anti-human IR800-coupled antibody (1:10,000; for all quantitation, detected by LiCor Odyssey). The presence of zSmo ectodomain protein was measured by colloidal Coomassie staining (GelCode Blue, Pierce/Thermo Scientific).

## Immunofluorescence to detect ciliary Smoothened

Cells were fixed with cold 4% paraformaldehyde (10 min, room temperature [RT]), washed with phosphate buffered saline (PBS; 3 times, 5 min each), placed in blocking solution (PBS, 1% (vol/vol) normal donkey serum, 0.1% (vol/vol) Triton X-100, 10 mg/ml bovine serum albumin) for 30 min at RT and then stained with primary antibodies (overnight, 4°C): anti-acetylated tubulin (#T6793; Sigma) at 1:3000 (vol/vol) dilution and anti-Smo (*Rohatgi et al., 2007*) at 1:500 (vol/vol) dilution in blocking solution. After washing (three times, 5 min in PBS + 0.1% Triton X-100), Alexa-coupled secondary antibodies (Jackson ImmunoResearch) were applied (1:500 [vol/vol] dilution, 1 hr, RT). Finally, stained cells were washed in PBS (three times, 5 min) and mounted onto glass slides with Prolong Gold mounting media with DAPI (Life Technologies).

## Microscopy and image analysis

The fixed cells were imaged with a Leica SP8 laser scanning confocal microscope, using a 63× oil objective (NA 1.40) and 1.3× zoom. For the quantitative analysis of Smo levels in cilia, all images used for comparisons were taken with identical gain, offset, and laser power settings on the microscope. Non-manipulated maximum projections of z-stacks were used for quantitation (Fiji). A mask, constructed by automatically applying a threshold to the acetylated tubulin image, was then applied to the corresponding anti-Smo image to measure Smo fluorescence at cilia. Local background correction was performed by moving the mask to measure fluorescence at a nearby region, and this value was subtracted from the ciliary Smo fluorescence.

## Data analysis

All statistical analysis and curve fitting were done in GraphPad Prism. For microscopy data, the Smo fluorescence for each cilium was individually plotted, generating a scatter plot that represents variability in the data. To compare Smo levels between different conditions, the median and interquartile range are provided (n = 60 for each condition).

For Hh reporter assays, each point is reported as the mean ± standard deviation (SD) derived from triplicates. Each result in the paper was repeated at least three times with similar outcomes. Relative luciferase activity was calculated by dividing Gli luciferase by Renilla luciferase luminescence. Fold-change in reporter activity was calculated by dividing each replicate by the mean reporter activity of the vehicle-treated control. Normalized (% of max) Hh reporter activity was calculated by setting the maximum value of a set to 100% and zero to 0% using the 'normalize' function of GraphPad Prism. In all graphs, dotted lines are straight connectors between points, and solid lines represent non-linear curve fits of the data (all done in GraphPad Prism). In *Figures 2C and 4E*, the curves were fit using the 'log(inhibitor) vs response—variable slope' function of GraphPad Prism. The model used for this function was $Y = Bottom + (Top–Bottom)/(1+10^{([LogIC50-X]*HillSlope)})$, where 'Y' represents bound zSmo as a percentage of the maximum bound (with zero competitor), 'Top' and 'Bottom' represent the plateaus at the beginning and end of the curve, respectively, and 'X' represents the concentration of free competitor added to the binding reaction. In *Figures 2B and 4D*, the curve was fit using the 'one site—total and nonspecific binding' function. The equation used for this fit incorporates both specific binding (specific = $Bmax*X/[X+Kd]$) and non-specific binding (nonspecific = $NS*X + Background$). 'X' in this case represents the sterol immobilized on the resin. In *Figure 8A,B,C*, the same 'log(inhibitor) vs response—variable slope' function as above was used to asses the IC50s of the OBIs in a Hh reporter assay.

## Crystallization and data collection

Prior to crystallization, the zSmo ectodomain from bacterial expression was concentrated to 7 mg/ml. Crystallization trials, using 100 nl protein solution plus 100 nl reservoir solution in sitting drop vapor diffusion format, were set up in 96-well Greiner plates using a Cartesian Technologies robot (*Walter*

*et al., 2005*). Crystallization plates were maintained at 20.5°C in a TAP Homebase storage vault and imaged via a Veeco visualization system (*Mayo et al., 2005*). zSmo ectodomain native and selenomethionine-substituted crystals were obtained out of mother liquor containing 100 mM HEPES pH7.0, PEG 6000 20%, 10 mM $ZnCl_2$.

X-ray diffraction data were collected at 100 K and crystals were treated with 25% (vol/vol) glycerol in mother liquor for cryo protection. Data were collected at beamline I03 at the Diamond Light Source, UK (native zSmo ectodomain), and at beamline ID23-EH1 (selenomethionine-substituted zSmo ectodomain) at the European Synchrotron Radiation Facility (ESRF), France. X-ray data were processed and scaled with the HKL suite (*Otwinowski and Minor, 1997*) and XIA2 (*Evans, 2006*; *Kabsch, 2010*; *Winter, 2010*). Data collection statistics are shown in *Table 1*.

## Structure determination, refinement and analyses of zSmo ectodomain

The zSmo ectodomain crystal structure was determined by single anomalous dispersion (SAD) analysis. The positions of three selenium atoms were determined using SHELXD (*Schneider and Sheldrick, 2002*). This solution was used as an input into the AutoSol module of the PHENIX suite (*Adams et al., 2002*) for phase calculation and improvement. The resulting map was of high quality and allowed tracing of the whole polypeptide chain (*Figure 5—figure supplement 1A*). An initial model was built automatically using RESOLVE (*Terwilliger, 2003*) and completed manually using COOT (*Emsley and Cowtan, 2004*). Iterative rounds of refinement in autoBUSTER (*Blanc et al., 2004*), PHENIX (*Adams et al., 2002*) and REFMAC (*Murshudov et al., 1997*) applying non-crystallographic symmetry restraints as well as manual building in COOT (*Emsley and Cowtan, 2004*) resulted in a well-defined model for zSmo ectodomain that included two molecules in the asymmetric unit both composed of residues 41–158 (*Figure 5—figure supplement 1B*). The zSmo ectodomain N- and C-terminal regions could not be traced due to missing electron density and were not included in the final model. The native structure was solved by molecular replacement using PHASER (*McCoy et al., 2007*) with the SeMet-labeled structure as a search model and refined as described above for the SeMet-labeled protein. Crystallographic and Ramachandran statistics are given in *Table 1*. Stereochemical properties were assessed by MolProbity (*Davis et al., 2007*). Superposition of CRD structures and root mean square deviation (RMSD) values were calculated for equivalent Cα atoms using program SHP (*Stuart et al., 1979*; *Riffel et al., 2002*). The phylogenetic tree for CRDs (*Figure 5B*) was prepared with program PHYLIP (*Felsenstein, 1989*) with the summed structural correlation data presented in *Figure 5—figure supplement 3* to construct a distance matrix. The program VOLUMES (RE Esnouf, unpublished) was used with a 1.4 Å radius probe to analyze the CRD binding grooves of zSmo and Fz8. The analysis of evolutionary conserved residues among the CRDs of the Smoothened family members was based on 80 amino acid sequences of vertebrate Smo CRDs and was mapped onto the zSmo CRD crystal structure using ProtSkin (*Deprez et al., 2005*).

## Molecular docking and homology modeling

The refined atomic coordinates of the zSmo CRD crystal structure were kept rigid during the molecular docking. The guanidinium group of Arg139 forms a hydrogen bond with a carbonyl oxygen of Arg139 in the neighboring molecule and occludes the oxysterol-binding pocket. Thus, the mmt180 rotamer (*Lovell et al., 2000*) of Arg139 was used during docking and pocket analysis. Atomic coordinates of 20(S)-OHC were downloaded from PubChem (compound ID 121935, *Wang et al., 2009*) and kept flexible during docking in AutoDock 4.2.5.1 using the Lamarckian genetic algorithm and default parameters (*Morris et al., 2009*). Estimated inhibition constant, $K_i$ (dissociation constant of the zSmo CRD-20(S)-OHC-complex), was calculated using formula $K_i = \exp(\Delta G/[R*T])$, where $\Delta G$ is a free energy of binding in kcal/mol, R is the gas constant 1.987 cal $K^{-1}$ $mol^{-1}$, and T = 298.15 K. The homology model of dSmo CRD (Ile82-Thr204, UniProtKB ID P91682) was built using program MODELLER 9.9 (*Eswar et al., 2008*) with the zSmo CRD structure as a template. The amino acid sequence identity between the corresponding CRD regions is 42%.

## Multi angle light scattering (MALS)

MALS analysis of purified and glycosylated zebrafish Smo ectodomain (expressed from mammalian cells) was performed using an analytical Superdex S200 10/30 size exclusion chromatography column (GE Heathcare) eluted in 150 mM NaCl, 10 mM HEPES pH 7.5 (flow rate 0.5 ml/min) with static light

scattering (DAWN HELEOS II, Wyatt Technology, Santa Barbara, CA), differential refractive index (Optilab rEX, Wyatt Technology) and Agilent 1200 UV (Agilent Technologies, Santa Clara, CA) detectors. Data were analyzed using the program ASTRA (Wyatt Technology).

## Zebrafish strain and husbandry
Adult fishes were maintained on a 14-hr light/10-hr dark cycle at 28°C in the AVA (Singapore) certified IMCB Zebrafish Facility. Zebrafish strain used was *Tg(eng2a:eGFP)*[i233] (*Maurya et al., 2011*).

## Zebrafish oxysterol treatment and in situ hybridization
The embryos of *Tg(eng2a:eGFP)*[i233] were dechorinated using pronase (Roche) at one cell stage. The well-developing ones at the 50% epiboly stage were selected and grown in fish water containing 50 µM 20(*S*)-OHC or 40 µM cyclopamine. Control embryos were kept in water containing the same amount of ethanol, used as the vehicle for the drugs. Standard in situ hybridization (ISH) was performed with anti-Dig alkaline phosphatase and chromogenic substrate NBT/BCIP as previously described (*Oxtoby and Jowett, 1993*). *ptch2* (formerly *ptc1*) RNA probe was prepared from template as previously described (*Concordet et al., 1996*).

## Chemical synthesis (detailed methods and characterization)
4-Bromo-1-trimethylsilyl-1-butyne; (1)

Prepared according to a known literature procedure (*Dieter and Chen, 2006*), CBr$_4$ (8.5 g, 25.6 mmol) was added to a solution of commercially available **4-trimethylsilyl-3-butyn-1-ol** (2.0 g, 14.06 mmol) in dry dichloromethane (DCM; 40 ml) at −30°C under N$_2$. The mixture was stirred vigorously for 10 min, until the CBr$_4$ was completely dissolved, whereupon a solution of PPh$_3$ (5.53 g, 21.09 mmol) in dry DCM (12 ml) was added dropwise. The reaction mixture was stirred at −30°C for 2 hr, after which the temperature was raised to 0°C and was allowed to slowly warm to RT over the next 2 hr. Upon completion, the reaction mixture was filtered through a pad of silica and concentrated in vacuo. The residue was purified by column chromatography on silica gel (100% hexane elution), to yield compound **1** as a colorless liquid (2.16 g, 75%). Analytical data are as previously reported (*Nachtergaele et al., 2012*).

(3β, 17β)-17-[(1*R,S*)-1-hydroxypent-4-yn-1-yl]-3-methoxymethoxyandrost-5-ene; (2)

Magnesium metal turnings (0.21 g, 8.66 mmol) in anhydrous Et$_2$O (15 ml) under N$_2$ were stirred in a two-necked flask equipped with a condenser. Compound **1** (1.77 g, 8.66 mmol) was added, followed by a few drops of 1,2-dibromoethane. The reaction mixture was warmed slightly to 30°C and stirred vigorously until

cloudiness was observed (~1–3 min). The reaction mixture was stirred an additional 30 min at RT, until the magnesium turnings were mostly consumed and the solution turned a murky yellow color. The flask was then cooled to 0°C, and a solution of **(3β, 17β)-3-methoxymethoxyandrost-5-ene-17-carboxaldehyde** (0.26 g, 0.75 mmol, *Nachtergaele et al., 2012*) in anhydrous THF (9 ml) was added dropwise to the reaction. After 10 min, two new spots were formed on TLC, indicating formation of a diastereomeric mixture, and the reaction was quenched with $NH_4Cl_{(aq)}$ (10 ml). The phases were separated, and the aqueous phase was extracted with $Et_2O$ (3 × 25 ml). The combined organic fractions were then washed with brine (1 × 25 ml), dried over $Na_2SO_4$, and concentrated in vacuo. The residue was quickly filtered through a small silica gel column (acetone–hexane, gradient elution). The product was concentrated, re-dissolved in anhydrous THF (15 ml) and cooled to 0°C. TBAF (0.96 ml, 1M in THF) was then added dropwise, and the reaction was allowed to stir for 10 min $H_2O$ (25 ml) was then added, and the reaction mixture was extracted with EtOAc (3 × 25 ml). The combined organic fractions were then dried over $Na_2SO_4$, and concentrated in vacuo. The compound was purified by column chromatography on silica gel (acetone–hexanes, gradient elution), to yield a diastereomeric mixture of compound **2** as a white solid, in 93% over two steps.

**Compound 2($R$)**: m.p. 135–137°C (EtOAc-hexanes); $[\alpha]_D^{20}$ =—24.0 ($c$ = 0.15, $CHCl_3$); IR: 3391, 2932, 1436, 1148, 1105, 1036 cm$^{-1}$; $^1$H NMR ($CDCl_3$) δ 5.29 (br s, 1H, H-6), 4.62 (s, 2H, OC$H_2$OCH$_3$), 3.55-3.66 (br m, 1H, HOC$H$), 3.30-3.40 (m, 1H, H-3), 3.30 (s, 3H, OCH$_2$OC$H_3$), 0.85-2.05 (m, 25H) 0.95 (s, 3H, H-19), 0.72 (s, 3H, H-18); $^{13}$C NMR ($CDCl_3$) δ 141.0, 121.8, 94.9, 77.1, 84.8, 73.7, 68.9, 56.6, 56.3, 55.4, 50.3, 42.6, 40.1, 39.8, 37.4, 37.0, 35.5, 32.1, 31.9, 29.1, 25.6, 24.8, 21.1, 19.6, 14.9, 12.6; HR-FAB MS [M+Na]$^+$ calculated for $C_{26}H_{40}O_3Na^+$: 423.2875, found 423.2878.

**Compound 2($S$)**: m.p. 111–113°C (EtOAc-hexanes); $[\alpha]_D^{20}$ =—48.7 ($c$ = 0.22 , $CHCl_3$); IR: 3270, 2933, 1628, 1437, 1148, 1101, 1040 cm$^{-1}$; $^1$H NMR ($CDCl_3$) δ 5.35 (br s, 1H, H-6), 4.68 (s, 2H, OC$H_2$OCH$_3$), 3.67-3.75 (br m, 1H, HOC$H$), 3.34-3.47 (m, 1H, H-3), 3.36 (s, 3H, OCH$_2$OC$H_3$), 0.85-2.40 (m, 25H) 1.00 (s, 3H, H-19), 0.71 (s, 3H, H-18); $^{13}$C NMR ($CDCl_3$) δ 140.9, 121.8, 94.9, 84.7, 77.1, 72.4, 68.7, 56.7, 56.5, 55.4, 50.3, 41.9, 39.7, 39.0, 37.4, 36.9, 35.5, 32.1, 31.7, 29.1, 25.3, 24.3, 21.0, 19.6, 15.1, 12.9; HR-FAB MS [M+Na]$^+$ calculated for $C_{26}H_{40}O_3Na^+$: 423.2875, found 423.2869.

## (3β, 17β)-17-(1-oxopent-4-yn-1-yl)-androst-5-en-3-ol; (20-keto-yne)

To a stirred solution of compound **2** (0.14 g, 0.34 mmol) in acetone (7 ml), freshly prepared Jones Reagent was added dropwise (~0.2 ml; 30% $CrO_3$·30% $H_2SO_4$·40% $H_2O$); adding until the reaction solution turned from green to yellow. Monitoring by TLC showed the two diastereomer spots corresponding to **2($R,S$)**, to be converted into one product spot. The reaction was subsequently quenched with isopropanol (3 ml), and the resulting blue-green solution was filtered through a pad of silica and then concentrated in vacuo. The resulting solid was then dissolved in MeOH:$H_2O$ (20 ml:10 drops), whereupon acetyl chloride (0.5 ml) was added dropwise over 15 min, and the reaction was stirred for 16 hr. Upon completion, the reaction flask was cooled to 0°C and was neutralized with sat. $NaHCO_{3(aq)}$ (~10 ml). The reaction mixture was extracted with DCM (4 × 10 ml), and the organic fractions were combined, dried over $Na_2SO_4$, and concentrated in vacuo. The residue was purified by column chromatography on silica gel (acetone–hexane, gradient elution), to yield compound **2** as a white solid in 97% over two steps.

**Compound 20-keto-yne**: m.p. 148–150°C (EtOAc-hexanes); $[\alpha]_D^{20}$ = +15.4 ($c$ = 0.19, $CHCl_3$); IR: 3294, 2928, 1704, 1435, 1382, 1097, 1047 cm$^{-1}$; $^1$H NMR ($CDCl_3$) δ 5.32 (br s, 1H, H-6), 3.45-3.58 (m, 1H, H-3) 0.96-2.64 (m, 25H) 0.98 (s, 3H, H-19), 0.60 (s, 3H, H-18); $^{13}$C NMR ($CDCl_3$) δ 209.2, 140.9, 121.4, 83.7, 71.7, 68.7, 62.9, 57.0, 50.0, 44.5, 43.0, 42.3, 39.0, 37.4, 36.6, 31.9, 31.0, 31.7, 24.6, 23.0, 21.2, 19.5, 13.5, 13.0; HR-FAB MS [M+Na]$^+$ calculated for $C_{24}H_{34}O_2Na^+$: 377.2457, found 377.2462.

## (3β, 17β)-17-[(2R)-2-hydroxy-hex-4-yn-2-yl]-androst-5-en-3-ol; (20(R)-yne)

Following a reported protocol (*Mijares et al., 1967*), **20-keto-yne** (78 mg, 0.22 mmol) was dissolved in anhydrous $Et_2O$ (10 ml) under $N_2$ and cooled to −25°C. Methylmagnesium bromide (3M in $Et_2O$; 0.37 ml, 1.10 mmol) was added dropwise, and the reaction mixture was allowed to warm to RT for 2 hr. The reaction was then diluted with $Et_2O$ (10 ml) and quenched with $NH_4Cl_{(aq)}$ (10 ml). The phases were separated, and the aqueous phase was extracted with $Et_2O$ (3 × 10 ml). The combined organic fractions were then washed with brine (1 × 15 ml), dried over $Na_2SO_4$, and concentrated in vacuo. The residue was purified by column chromatography on silica gel (EtOAc–toluene, gradient elution), to yield the desired **20(R)-yne** as a white solid in 85% yield.

Compound **20(R)-yne**: m.p. 154–157°C (EtOAc-hexanes); $[\alpha]_D^{20}$ =—58.2 (c = 0.18 , $CHCl_3$); IR: 3307, 2928, 1645, 1433, 1381, 1054 cm$^{-1}$; $^1$H NMR ($CDCl_3$) δ 5.35 (d, 1H, $^3J$ = 5.6 Hz, H-6), 3.44-3.57 (m, 1H, H-3), 1.14 (s, 3H, H-21), 1.01 (s, 3H, H-19), 0.87 (s, 3H, H-18); $^{13}$C NMR ($CDCl_3$) δ 141.0, 121.7, 85.3, 75.5, 71.9, 68.6, 59.1, 57.0, 50.2, 43.1, 42.4, 41.1, 40.4, 37.4, 36.7, 31.9, 31.8, 31.5, 26.3, 24.0, 23.3, 21.1, 19.6, 13.9, 13.5; HR-FAB MS [M+Na]$^+$ calculated for $C_{25}H_{38}O_2Na^+$: 393.2770, found 393.2771.

## (3β, 17β)-17-[(2S)-2-hydroxypent-4-ene-2-yl]-androst-5-en-3-ol; (3)

Compound **3** was made using Barbier reaction conditions (*Barbier, 1899*) (one-pot Gringard reaction) following the procedure of *Shaw (1966)*. A portion of the magnesium turnings (2.92 g, 120.2 mmol) and a portion of the total allyl bromide (2.95 ml, 34.1 mmol) were stirred in a flame-dried 3-neck flask, fitted with a reflux condenser and a dropping funnel, under $N_2$, containing in anhydrous $Et_2O$ (100 ml). The reaction mixture was warmed slightly (30°C), a few drops of 1,2-dibromoethane were added and the reaction mixture was stirred vigorously for 5 min, or until the solution appeared cloudy. The heat source was removed, and a solution of pregnenolone (3.0 g, 9.48 mmol) and allyl bromide (7.0 ml, 81.5 mmol) in $Et_2O$:THF (1:2, vol/vol; 230 ml) was added to the dropping funnel. A quarter of the steroid mixture (~60 ml) was slowly added over 7 min, at which time, another portion of magnesium turnings was added (0.23 g, 9.5 mmol). This sequence of steroid addition followed by magnesium addition was repeated thrice. The mixture was then refluxed for 2 hr at 40°C. Upon completion, the reaction mixture was diluted with $Et_2O$ (50 ml) and quenched slowly with $NH_4Cl_{(aq)}$ (50 ml). The reaction mixture was filtered through cotton, to remove any remaining magnesium turnings, and into a separatory funnel, where upon additional $H_2O$ (100 ml) was added. The organic layer was then extracted with $Et_2O$ (3 × 1000 ml), dried over $Na_2SO_4$, and concentrated in vacuo. The residue was purified by column chromatography on silica gel (EtOAc–hexanes, gradient elution), to yield compound **3** as a mixture of diastereomers (*R:S*; 1:9) as a white solid in 98% yield. The desired **3(S)** diastereomer was obtained through recrystallization from DCM:MeOH. Spectral data were as previously reported (*Colonna and Gros, 1973*).

## (3β, 17β)-3-methoxymethoxy-17-[(2S)-2-methoxymethoxy-pent-4-en-2-yl]-androst-5-ene; (4)

Compound **3** (3.37 g, 9.40 mmol) was dissolved in dry DCM (150 ml) in a 2-neck flask fitted with a reflux condenser at 0°C. Diisopropyl ethylamine (13.0 ml, 75.19 mmol) and a catalytic amount of DMAP (57 mg) were next added to the flask. MOMCl (2.86 ml, 37.59 mmol) was then added dropwise, and after 10 min, the reaction mixture was removed from the ice bath and was refluxed at 50° for 24 hr. Upon completion, the reaction was quenched with $H_2O$ (25 ml). The reaction mixture was then washed sequentially with sat. $NaHCO_{3(aq)}$ (50 ml) and $NaCl_{(aq)}$ (50 ml), dried over $Na_2SO_4$, and concentrated in vacuo. The residue was purified by column chromatography on silica gel (EtOAc–hexanes, gradient elution), to yield the desired compound **4** as a white solid in 95% yield.

**Compound 4**: m.p. 87–88°C (MeOH-DCM); IR: 2935, 2891, 2846, 1639, 1464, 1439, 1371, 1147, 1106, 1037, 916 cm$^{-1}$; $^1$H NMR (CDCl$_3$) δ 5.62-5.78 (m, 1H, C$H$=CH$_2$), 5.30 (br d, 1H, $^3J$ = 5.6 Hz, H-6), 4.93-5.45 (2 d, 2H, CH=C$H_2$) 4.71 (dd, 1H, $J$ = 1.2 Hz, $^3J$ = 9.8 Hz C(17)-COC$H_2$OCH$_3$), 4.62 (s, 2H, C(3)-OC$H_2$OCH$_3$), 4.61 (dd, 1H, $J$ = 1.2 Hz, $^3J$ = 9.8 Hz, C(17)-COC$H_2$OCH$_3$), 3.30-3.42 (m, 1H, H-3), 3.31 (2 s, 6H, C(17)-COCH$_2$OC$H_3$, C(3)-COCH$_2$OC$H_3$), 1.28 (s, 3H, C(17)-CC$H_3$), 0.96 (s, 1H, H-19), 0.80 (s, 3H, H-18); $^{13}$C NMR (CDCl$_3$) δ 140.8, 135.1, 121.7, 117.2, 94.7, 90.7, 80.0, 77.0, 57.4, 56.9, 55.5, 55.1, 50.2, 45.4, 42.6, 40.4, 39.6, 37.3, 36.8, 31.9, 31.4, 29.0, 23.9, 23.0, 22.1, 21.0, 19.4, 13.9; Anal. Calculated. for $C_{28}H_{46}O_4$: C, 75.29; H, 10.38. Observed: C, 75.19; H, 10.48.

## (4S)-[4-[(3β, 17β)-3-methoxymethoxyandrost-5-en-17-yl]-4-methoxymethoxy]-pentan-1-ol; (5)

Compound **4** (2.5 g, 5.6 mmol) was dissolved in THF (50 ml) under $N_2$, and 9-BBN (56 ml, 0.5M in THF) was added dropwise. The reaction mixture was allowed to stir at RT for 16 hr, and upon reaction completion [3N] NaOH$_{(aq)}$ (8 ml) was carefully added, followed by the careful addition of 30% $H_2O_2$ (8 ml), and the reaction was stirred for 1 hr. $H_2O$ was added (50 ml), and the reaction mixture was extracted with EtOAc (3 × 50 ml), dried over $Na_2SO_4$, and concentrated in vacuo. The residue was purified by column chromatography on silica gel (EtOAc–hexanes, gradient elution), to yield the desired compound **5** as a white solid in 96% yield.

**Compound 5**: m.p. 87–90°C (cold MeOH); IR: 3436, 2934, 2899, 2848, 1464, 1439, 1370, 1147, 1105, 1036, 916 cm$^{-1}$; $^1$H NMR (CDCl$_3$) δ 5.32 (br d, 1H, $^3J$ = 6.8 Hz, H-6), 4.70 (d, 1H, $^3J$ = 9.6 Hz, C(17)-COC$H_2$OCH$_3$), 4.65 (s, 2H, C(3)-OC$H_2$OCH$_3$), 4.63 (d, 1H, $^3J$ = 9.6 Hz, C(17)-COC$H_2$OCH$_3$), 3.52-3.61 (m, 2H, C$H_2$OH), 3.32-3.44 (m, 1H, H-3), 3.33 (2 s, 6H, C(17)-COCH$_2$OC$H_3$, C(3)-COCH$_2$OC$H_3$), 1.27 (s, 3H, C(17)-CC$H_3$), 0.98 (s, 1H, H-19), 0.80 (s, 3H, H-18); $^{13}$C NMR (CDCl$_3$) δ 140.8, 121.8, 94.8, 90.6, 80.6, 77.1, 63.3, 57.3, 57.1, 55.6, 55.3, 50.2, 42.6, 40.4, 39.7, 37.4, 36.8, 36.3, 31.9, 31.5, 29.0, 28.2, 23.9, 23.0, 22.7, 21.0, 19.4, 14.1; Anal. calculated for $C_{28}H_{48}O_5$: C, 72.37; H, 10.41. Observed: C, 72.60; H, 10.16.

## (3β, 17β)-3-methoxymethoxy-17-[(2S)-5-azido-2-methoxymethoxy-pentan-2-yl]-androst-5-ene; (6)

In a similar manner to reported protocol (*Pore et al., 2006*), compound **5** (1.51 g, 3.25 mmol) was dissolved in dry DCM (20 ml) under $N_2$. The reaction mixture was cooled to 0°C, and $Et_3N$ (0.91 ml, 6.5 mmol) was added. A solution of methanesulfonyl chloride (0.38 ml, 4.87 mmol) in dry DCM (7 ml) was added dropwise to the reaction over 10 min. The reaction was allowed to stir for 30 min at 0°C. Upon reaction completion, ice water (10 ml) was added. The organic phase was separated and sequentially washed with sat. $NaHCO_{3(aq)}$ (10 ml), brine (10 ml), dried over $Na_2SO_4$, and concentrated in vacuo; taking care not to warm the flask over 30°C in the process. The crude mesylated steroid was allowed to dry under vacuum for 2–3 hr, upon which time it was transferred to a 2-neck flask fitted with a reflux condenser, and was re-dissolved in anhydrous DMF (50 ml). $NaN_3$ (1.06 g, 16.25 mmol) was then added, and the reaction was stirred at 65°C for 1 hr. The reaction was then dumped into a beaker of ice water (50 ml) to quench the $NaN_3$, transferred to a separatory funnel and extracted with $EtOAc$:$Et_2O$ (1:2, vol/vol; 4 × 25 ml). The organic fractions were combined and dried over $Na_2SO_4$ and concentrated in vacuo. The residue was purified by column chromatography on silica gel (EtOAc–hexanes, gradient elution), to yield the desired compound **6** as a white solid in 94% yield.

**Compound 6**: m.p. 78–79°C (EtOAc:hexanes); IR: 2963, 2934, 2895, 2833, 2094, 1462, 1437, 1371, 1265, 1140, 1106, 1090, 1041, 914 cm⁻¹; ¹H NMR (CDCl₃) δ 5.33 (br d, 1H, ³J = 5.6 Hz, H-6), 4.73 (dd, 1H, J = 1.6 Hz, ³J = 9.4 Hz, C(17)-COCH₂OCH₃), 4.66 (d, 2H, J = 1.6 Hz, C(3)-OCH₂OCH₃), 4.62 (dd, 1H, J = 1.6 Hz, ³J = 9.4 Hz, C(17)-COCH₂OCH₃), 3.34-3.46 (m, 1H, H-3), 3.34 (2 s, 6H, C(17)-COCH₂OCH₃, C(3)-COCH₂OCH₃), 3.18-3.27 (m, 2H, CH₂N₃), 1.29 (s, 3H, C(17)-CCH₃), 0.99 (s, 1H, H-19), 0.81 (s, 3H, H-18); ¹³C NMR (CDCl₃) δ 140.9, 121.8, 94.8, 90.7, 80.3, 77.1, 57.4, 57.1, 55.6, 55.3, 52.2, 50.3, 42.7, 40.4, 39.7, 37.4 (x2), 36.9, 32.0, 31.5, 29.1, 24.6, 23.9, 23.0, 22.7, 21.0, 19.5, 14.1; Anal. calculated for $C_{28}H_{47}N_3O_4$: C, 68.68; H, 9.67; N, 8.58. Observed: C, 68.47; H, 9.48; N, 8.59.

## (3β, 17β)-17-[(2S)-5-azido-2-hydroxy-pentan-2-yl]-androst-5-en-3-ol; (7)

Compound **6** (750 mg, 1.53 mmol) was dissolved in MeOH:DCM (80 ml:20 ml) and cooled to 0°C. AcCl (2.5 ml) was added dropwise, and the reaction was kept between 0°C and 10°C to minimize competing elimination byproducts. After 2 days, the starting material was found to be completely consumed, and the reaction mixture was cooled down to 0°C, and quenched very slowly with sat. $NaHCO_{3(aq)}$. The reaction was then diluted further with $H_2O$ and extracted with DCM (4 × 25 ml). The combined organic fractions were then dried over $Na_2SO_4$ and concentrated in vacuo. The residue was purified by column chromatography on silica gel (EtOAc–hexanes, gradient elution), to yield the desired compound **7** as a white solid in 65% yield.

**Compound 7**: m.p. 106–108°C (EtOAc:hexanes); [α]$_D^{20}$ =—52.87 (c = 0.23 , CHCl₃); IR: 3400, 2935, 2902, 2868, 2097, 1464, 1376, 1353, 1260, 1055 cm⁻¹; ¹H NMR (CDCl₃) δ 5.28 (br d, 1H, ³J = 5.6 Hz,

H-6), 3.36-3.52 (m, 1H, H-3), 3.11-3.28 (m, 2H, C$H_2$N$_3$), 1.21 (s, 3H, H-21), 0.93 (s, 3H, H-19), 0.79 (s, 3H, H-18); $^{13}$C NMR (CDCl$_3$) δ 141.0, 121.7, 75.1, 71.9, 58.4, 57.1, 52.2, 50.2, 42.9, 42.5, 40.6, 40.3, 37.4, 36.7, 31.9, 31.8, 31.5, 26.4, 24.0 (x2), 22.6, 21.1, 19.6, 13.7; Anal. Calculated for C$_{24}$H$_{39}$N$_3$O$_2$: C, 71.78; H, 9.79; N, 10.46. Observed: C, 71.68; H, 9.66; N, 10.27.

### (3β, 17β)-17-[(2S)-5-amino-2-hydroxy-pentan-2-yl]-androst-5-en-3-ol; (20(S)-amine)

In a similar manner to reported protocols (*Zhao and Zhong, 2005*; *Ryu et al., 2006*), compound **7** (150 mg, 0.374 mmol) was dissolved in THF:H$_2$O (2.0 ml: 0.05 ml) and warmed to 30°C for 30 min. The reaction was then allowed to cool to room temperature and stirred for an additional 24 hr. The reaction was monitored by TLC (MeOH:EtOAC:Et$_3$N, 1:1:0.1), and upon completion, the solvents were evaporated, keeping water bath below 35°C. The residue was then redissolved in a minimal amount of MeOH:DCM and purified by column chromatography on a silica gel column packed with hexanes:Et$_3$N (49:1). Gradient elution was started using EtOAc-hexanes (adding ~0.5% Et$_3$N to each gradient) to remove PPh$_3$ byproducts; then transitioning to 100% DCM (still adding 0.5% Et$_3$N); and the compound was finally eluted by flushing with 1% MeOH-DCM (still adding 0.5% Et$_3$N) to give compound **20(S)-amine** in 90% yield as a white solid, without the need for further purification. Attempts to recrystallize the product from MeOH were unsuccessful, and prolonged periods in the solvent seemed induce the formation of a new spot on TLC, presumably an ammonium salt. Thus, the compound was best purified only by column chromatography.

**Compound 20(S)-amine**: m.p. 167–172°C (amorphous solid; MeOH:DCM); IR: 3368, 2931, 2900, cm$^{-1}$; $^1$H NMR (CDCl$_3$:MeOD; ~20:1) δ 5.35 (d, 1H, $^3J$ = 7.2 Hz, 1H, H-6), 3.46-3.59 (m, 1H, H-3), 2.70-2.82 (m, 1H, C$H_2$NH$_2$), 2.58-2.70 (m, 1H, C$H_2$NH$_2$), 2.17-2.35 (m, 2H), 2.06-2.15 (dt, 1H), 1.93-2.04 (m, 1H), 1.28 (s, 3H, H-21), 1.02 (s, 3H, H-19), 0.87 (s, 3H, H-18); $^{13}$C NMR (CDCl$_3$:MeOD; ~20:1) δ 141.0, 121.4, 74.7, 71.3, 58.4, 57.0, 50.1, 42.6, 42.1, 42.0, 40.7, 40.2, 37.3, 36.5, 31.8, 31.3, 31.3, 27.4, 25.7, 23.8, 22.4, 20.9, 19.3, 13.4; HR-FAB MS [M+H]$^+$ calculated for C$_{24}$H$_{42}$NO$_2{}^+$: 376.3216, found 376.3208.

## Acknowledgements

We thank the staff of beamline I03 from the Diamond Light Source, UK, and beamline ID23-EH1 from the European Synchrotron Radiation Facility, France, for assistance with data collection, B Bishop for laboratory support, RM Esnouf for help with volume calculations, and Luigi De Colibus, Alex Evers, Pehr Harbury and Suzanne Pfeffer for helpful discussions. We thank C Hughes, G Pusapati, G Luchetti, and A Lebensohn for help with experiments and P Lovelace for help with FACS. Mass spectrometry analysis was conducted at the NIH/NCRR Mass Spectrometry Facility at Washington University, supported by the NIH (RR00954, DK020579, DK056341). The crystal structures presented in this work have been deposited under PDB codes 4C79 and 4C7A.

## Additional information

### Funding

| Funder | Grant reference number | Author |
| --- | --- | --- |
| National Institutes of Health | GM47969, HL67773 | Kathiresan Krishnan, Douglas F Covey |
| National Institutes of Health | 5 T32 HL007275 | Laurel K Mydock |
| Stand up to Cancer Foundation | SU2C-AACR-IRG0209 | Sigrid Nachtergaele, Rajat Rohatgi |

| Funder | Grant reference number | Author |
|---|---|---|
| Pew Scholars in the Biomedical Sciences | 2009-000359-011 | Sigrid Nachtergaele, Rajat Rohatgi |
| Wellcome Trust | WT/082301/Z/07/Z | Christian Siebold |
| Cancer Research UK | C20724/A14414 | Christian Siebold |
| A*STAR Singapore | | Zhonghua Zhao, Philip W Ingham |
| National Science Foundation | | Sigrid Nachtergaele |
| Medical Research Council | | Daniel M Whalen, Tomas Malinauskas |

The funders had no role in study design, data collection and interpretation, or the decision to submit the work for publication.

## Author contributions

SN, DMW, LKM, TM, PWI, DFC, CS, RR, Conception and design, Acquisition of data, Analysis and interpretation of data, Drafting or revising the article, Contributed unpublished essential data or reagents; ZZ, Acquisition of data, Analysis and interpretation of data, Drafting or revising the article, Contributed unpublished essential data or reagents; KK, Acquisition of data, Drafting or revising the article, Contributed unpublished essential data or reagents

## Ethics

Animal experimentation: Zebrafish were maintained in a facility accredited by the Association for Assessment and Accreditation of Laboratory Animal Care International, inspected annually by Agri-Food and Veterinary Authority of Singapore and quarterly by Biological Resource Centre to ensure strict adherence to the stipulated animal welfare guidelines. All animals were handled according to approved institutional care and use committee (IUCAC) protocols (#110638 and #120751) of the National University of Singapore.

# Additional files

## Major dataset

The following datasets were generated:

| Author(s) | Year | Dataset title | Dataset ID and/or URL | Database, license, and accessibility information |
|---|---|---|---|---|
| Nachtergaele S, Whalen DM, Mydock LK, Zhao Z, Malinauskas T, Krishnan K, et al. | 2013 | Crystal structure of the Smoothened CRD, native | 4C79; http://www.rcsb.org/pdb/search/structidSearch.do?structureId=4C79 | Publicly available at the RCSB Protein Data Bank (http://www.rcsb.org/). |
| Nachtergaele S, Whalen DM, Mydock LK, Zhao Z, Malinauskas T, Krishnan K, et al. | 2013 | Crystal structure of the Smoothened CRD, selenomethionine-labeled | 4C7A; http://www.rcsb.org/pdb/search/structidSearch.do?structureId=4C7A | Publicly available at the RCSB Protein Data Bank (http://www.rcsb.org/). |

The following previously published datasets were used:

| Author(s) | Year | Dataset title | Dataset ID and/or URL | Database, license, and accessibility information |
|---|---|---|---|---|
| Dann CE, Hsieh JC, Rattner A, Sharma D, Nathans J, Leahy DJ | 2001 | Crystal structure of the cysteine-rich domain of secreted Frizzled-related protein 3 (SFRP-3;FZB) | 1IJX; http://www.rcsb.org/pdb/explore/explore.do?structureId=1ijx | Publicly available at the RCSB Protein Data Bank (http://www.rcsb.org/). |
| Janda CY, Waghray D, Levin AM, Thomas C, Garcia KC | 2012 | Crystal structure of XWnt8 in complex with the cysteine-rich domain of Frizzled 8 | 4F0A; http://www.rcsb.org/pdb/explore/explore.do?structureId=4f0a | Publicly available at the RCSB Protein Data Bank (http://www.rcsb.org/). |

| | | | | |
|---|---|---|---|---|
| Stiegler AL, Burden SJ, Hubbard SR | 2009 | Crystal Structure of the Frizzled-like Cysteine-rich Domain of MuSK | 3HKL; http://www.rcsb.org/pdb/explore/explore.do?structureId=3hkl | Publicly available at the RCSB Protein Data Bank (http://www.rcsb.org/). |
| Kwon HJ, Abi-Mosleh L, Wang ML, Deisenhofer J, Goldstein JL, Brown MS, et al. | 2009 | NPC1(NTD):cholesterol | 3GKI; http://www.rcsb.org/pdb/explore/explore.do?structureId=3gki | Publicly available at the RCSB Protein Data Bank (http://www.rcsb.org/). |
| Monaco HL | 1997 | Crystal structure of the chicken riboflavin-binding protein. RFBP (this older dataset is not available in the RCSB Protein Databank) | | Available by email from Prof. Hugo L. Monaco (monaco@sci.univr.it), University of Verona, Italy. |
| Chen C, Ke J, Zhou XE, Yi W, Brunzelle JS, Li J, et al. | 2013 | Crystal structure of human folate receptor alpha in complex with folic acid | 4LRH; http://www.rcsb.org/pdb/explore/explore.do?structureId=4lrh | Publicly available at the RCSB Protein Data Bank (http://www.rcsb.org/). |

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
