## [Decision Letter]

Thank you for sending your work entitled “Structure and function of the Smoothened extracellular domain in vertebrate Hedgehog signaling” for consideration at *eLife*. Your article has been favorably evaluated by a Reviewing editor and 2 reviewers.

The Reviewing editor and the two reviewers discussed their comments before we reached this decision, and the Reviewing editor has assembled the following comments to help you prepare a revised submission. We anticipate that it will be straightforward to respond to these comments in a revised manuscript and, assuming that you are able to do so, you will receive a quick final decision from the editor without involving the reviewers any further.

The Hedgehog signaling pathway is important for tissue development and injury responses, and misregulation of this pathway can lead to cancer. Nachtergaele et al. build on previous work from their and other laboratories showing the ability of specific hydroxysterols to activate the Hedgehog signaling pathway through interactions with the 7TM protein Smoothened (Smo). In this new work, Nachtergaele and colleagues show that (i) hydroxysterols interact specifically with the N-terminal, extracellular cysteine-rich domain (CRD) of Smo (and their binding site is thus distinct from the well-characterized antagonist/agonist binding site in the 7 TM region), (ii) hydroxysterols interact with the Smo CRD at a site that homologous to the site on Frizzled CRDs that interacts with the palmitoleyl group attached to Wnt, (iii) hydroxysterol analogs can be created that show partial inhibition of Hedgehog signaling and hold promise as Hedgehog pathway-targeting therapeutics, (iv) interactions between Smo CRDs and hydroxysterols appear to be specific to vertebrate Smos, (v) hydroxysterols can activate Hedgehog signaling in fish embryos, and (vi) determine the 2.3 Å crystal structure of the Smo CRD from zebrafish.

These results are persuasive, well described, and represent significant advances in our understanding of how small molecules regulate Smo behavior. The Salic group recently published similar results in Nature Chemical Biology and we are aware of a paper in press at Developmental Cell also reporting similar results. There is overlap in many of the experiments and conclusions mapping the oxysterol binding domains of vertebrate Smoothened, the relationship between oxysterol engagement here and the action of inhibitors and activators within the 7 TM domain, and the reasons why *Drosophila* Smoothened works differently. Actual structural data on the CRD for Smoothened rather than inferred structural information in the Nedelcu based on lipid interactions with related CRDs for Frizzled family members distinguishes the paper under review. However, an actual structure with the CRD bound to oxysterol is not presented. Nevertheless, that all of these authors come to similar conclusions strengthens the impact of their work, and given the importance and interest in this molecule and the pathway it regulates, having more than one excellent report on this topic is no bad thing. Further, there are interesting differences in the behavior of “intelligent” drugs developed from an oxysterol base that adds an intriguing element to the current work.

Please address the following issues in a revised manuscript:

1) The authors discuss the possibility that cyclopamine may engage both the CRD and 7 TM domain, explaining properties such as cyclopamine's ability to induce cilial trafficking while non-sterol pathway inhibitors thought to act at the 7 TM region do not have this effect. However, cyclopamine does not appear to compete with oxysterol binding to the CRD region in the authors' biochemical studies—a result at odds with this conclusion (Figure 7). These data create confusion regarding the interplay the 7 TM and CRD sites). There are also recent data suggesting glucocorticoid interactions from Wang et al. that the authors might bring into the discussion. The authors are encouraged to open up some more in their discussion, making it very clear where their data agrees or disagrees with conclusions in Nedelcu et al., how they interpret disagreement and how they perceive the role of oxysterols in normal HH signaling.

2) There is an odd discrepancy in the figures that the authors should address. Firstly, in Figure 1 the Δ-CRD form of mSmo migrates higher on the gel than Wild-type mSmo when deletion of a domain suggests it should run lower (as it does in Figure 6). Is there some mistake in the labeling or loading of the gel in Figure 1? Also, in Figure 6 the blot is labeled anti-Fc, but these forms of mSmo do not appear to have been Fc-tagged (vs YFP tagged).

3) The description of the zebrafish embryo experiments (Figure 4) is difficult to follow for a more general audience. Please provide a more complete explanation and motivation for this experiment. Because only one example of each condition is shown and there is no data quantitation, it is difficult to know if the control and oxysterol treated samples are significantly different.

4) It would be useful to describe the fitting procedure for the bead-binding/competition curves and include a table that outlines the relative affinities for the different constructs/compounds studied.

5) The MALS experiments were performed at a zSmo CRD concentration of 5 μM. This is a very low concentration, so it is difficult to definitively state that the protein is monomeric. MALS experiments at higher protein concentrations would be useful to support the authors’ conclusion, or the authors could just say that the protein does not dimerize at low concentration.

6) A description of the structural comparison of the zSmo CRD is lacking. It is unclear what methods the authors used to make these comparisons and generate a phylogenetic tree. Are there other uncharacterized receptors that belong to this family? What are the rms deviations for the core parts of the structure, and over how many residues?

---

## [Author Response]

*1) The authors discuss the possibility that cyclopamine may engage both the CRD and 7 TM domain, explaining properties such as cyclopamine's ability to induce cilial trafficking while non-sterol pathway inhibitors thought to act atthe 7 TM region do not have this effect. However, cyclopamine does not appear to compete with oxysterol binding to the CRD region in the authors' biochemical studies—a result at odds with this conclusion (*Figure 7*). These data create confusion regarding the interplay the 7 TM and CRD sites*.

We thank the reviewers for pointing out the potential contradiction between the results presented Figures 7 and 8. Figure 8 is a ligand affinity chromatography assay demonstrating that cyclopamine can indeed inhibit binding of the isolated Smo CRD-Fc to 20(S)-OHC beads. This observation is the basis of our hypothesis that cyclopamine can engage both the CRD and the 7 TM domains. Figure 7 is an assay that measures binding of bodipy-conjugated cyclopamine to cells expressing full-length Smo. In this assay (as shown previously) 20(S)-OHC does not block binding of bodipy-cyclopamine to Smo-expressing cells. Thus, we detect competition between 20(S)-OHC and cyclopamine when using the isolated, purified Smo CRD but not when using cells expressing full-length Smo. We believe the difference is because the cell-binding assay is significantly less sensitive than the affinity chromatography assay and is largely reporting on the higher affinity interaction between cyclopamine and the 7 TM site (which cannot be competed with oxysterols). Alternatively, it is possible that cyclopamine binds the CRD more weakly when it is embedded in the context of the whole protein.

The distinction between binding to the isolated CRD protein (Figure 8) and full-length Smo (Figure 7) in cells is now explicitly described in the Discussion, along with the potential explanations mentioned above. In terms of the interplay between the two sites, we describe the CRD and 7 TM sites as being physically distinct but allosterically linked early in the discussion.

*There are also recent data suggesting glucocorticoid interactions from Wang et al that the authors might bring into the discussion*.

We agree that the Wang et al. paper is indeed quite relevant as it shows that Hh-active glucocorticoids fall into two distinct classes—a set that competes with cyclopamine and a set that does not—supporting the view that there are two ligand binding sites on Smo. This paper is now discussed and referenced both in the Results (in the context of our discussion on the oxysterol-based inhibitors) and in the Discussion.

*The authors are encouraged to open up some more in their discussion, making it very clear where their data agrees or disagrees with conclusions in Nedelcu et al., how they interpret disagreement and how they perceive the role of oxysterols in normal HH signaling*.

As described in the response to point 1, there are two distinct assays that have been used by us and by Nedelcu et al. to measure competition—binding of the isolated CRD to oxysterol beads and binding of bodipy cyclopamine to Smo expressing cells. Our data (Figure 7) are in agreement with Nedelcu et al. (and previous studies described in [24]) that oxysterols do not compete with cyclopamine for binding to Smo-expressing cells. Nedelcu et al. did not report on whether cyclopamine competes with 20(S)-OHC for binding to the isolated Smo CRD—the experiment we show in Figure 8—and thus there is no disagreement on this point.

We also now explicitly discuss (both in the Results and the Discussion) the following points of comparison between our manuscript and that of Nedelcu et al.:

A) The different properties of our CRD-targeted partial agonists of Smo compared to the CRD-targeted inhibitor from Nedelcu et al.

B) A comparison of Shh-responsiveness of the CRD-deleted Smo proteins reported in both the studies.

C) As the reviewers recommend, we have expanded the discussion to clarify our views on whether endogenous oxysterols regulate Hh signaling, whether oxysterols mediate the interaction between Ptch1 and Smo, and what specific studies will be required in the future to conclusively answer this important question.

*2) There is an odd discrepancy in the figures that the authors should address. Firstly, in*
Figure 1
*the Δ-CRD form of mSmo migrates higher on the gel than Wild-type mSmo when deletion of a domain suggests it should run lower (as it does in*
Figure 6*). Is there some mistake in the labeling or loading of the gel in*
Figure 1*? Also, in*
Figure 6
*the blot is labeled anti-Fc, but these forms of mSmo do not appear to have been Fc-tagged (vs YFP tagged)*.

We consistently find that YFP-Smo and ΔCRD-YFP-Smo migrate at roughly the same position when fractionated on 8% (37.5:1 Acrylamide:Bis) SDS-PAGE gels run using an alkaline (pH=8.8) Tris-Glycine buffer system. This is seen in Figure 1 and again in Figure 3, which depicts lysates from stable cell lines expressing YFP-Smo and ΔCRD-YFP-Smo. However, ΔCRD-YFP-Smo runs lower than YFP-Smo (as expected) when fractionated on a 4–12% gradient gel run using a neutral (Bis-Tris, pH=7.0) buffer system, which is used in Figure 6. Thus, the gel matrix composition and buffer system used had a large effect on the relative migration of YFP-Smo and ΔCRD-YFP-Smo. The different gel electrophoresis conditions are now noted in the respective figure legends.

Figure 6E is indeed an anti-YFP immunoblot (and not an anti-Fc immunoblot). We apologize for this error and have corrected it in the revised manuscript.

*3) The description of the zebrafish embryo experiments (*Figure 4*) is difficult to follow for a more general audience. Please provide a more complete explanation and motivation for this experiment. Because only one example of each condition is shown and there is no data quantitation, it is difficult to know if the control and oxysterol treated samples are significantly different*.

We have now expanded the description of the experiment and its motivation in the Results. The main motivation for the experiment is to demonstrate that zebrafish Smo (whose CRD we have crystallized and characterized in biochemical assays) does in fact respond to oxysterols. Since full length zebrafish Smo is poorly expressed in mammalian cells, we treated embryos to determine if endogenous Smo can respond to oxysterols by activating a transgenic Hh pathway reporter used to study Hh signaling during zebrafish development. In addition, data quantitation and statistical analysis of significance for this experiment is now provided in a new figure (Figure 4–figure supplement 1).

*4) It would be useful to describe the fitting procedure for the bead-binding/competition curves and include a table that outlines the relative affinities for the different constructs/compounds studied*.

The fitting procedures and equations for all curves, shown in both the binding and signaling assays, are now included in the Methods section. For the binding experiments, two proteins were used—the mouse Smo CRD-Fc and the zebrafish Smo ECD. We have included saturation curves for the binding of both proteins to 20(S)-OHC beads in Figure 2 (new in the revised manuscript) and 4D, respectively. Estimates of the dissociation constants for the binding of these two proteins to 20(S)-OHC beads are noted in the respective figure legends. We have not used competition curves to derive dissociation constants because we measured the ability of free ligand to block binding of the Smo CRD to bead immobilized ligand—a condition in which the two ligands are clearly chemically different (hence violating a key assumption when using competition to derive dissociation constants). Finally, Figure 8–figure supplement 1 has been added to summarize the IC50 values for the two oxysterol-based inhibitors, 20(R)-yne and 20-keto-yne.

*5) The MALS experiments were performed at a zSmo CRD concentration of 5 μM. This is a very low concentration, so it is difficult to definitively state that the protein is monomeric. MALS experiments at higher protein concentrations would be useful to support the authors’ conclusion, or the authors could just say that the protein does not dimerize at low concentration*.

We have clarified the text as suggested and changed “In agreement with this crystal packing analysis, purified zSmo ectodomain behaved as a monomer in solution at concentrations up to 5 μM when assessed using multi angle light scattering (Figure 5–figure supplement 1D)” to “In agreement with this crystal packing analysis, purified zSmo ectodomain behaved as a monomer in solution at low concentrations (5 μM) when assessed using multi angle light scattering (Figure 5–figure supplement 1D).”

*6) A description of the structural comparison of the zSmo CRD is lacking. It is unclear what methods the authors used to make these comparisons and generate a phylogenetic tree*.

We apologize for not being clearer in our description of the structural phylogenetic tree analysis. We have now included a description in the Methods and replaced “Structure-based phylogenetic analysis of the CRD domains was performed using SHP (62)”with “Superposition of CRD structures and root mean square deviation (RMSD) values were calculated for equivalent Cα atoms using program SHP (76; 62). The phylogenetic tree for CRDs (Figure 5) was prepared with program PHYLIP (28) with the summed structural correlation data presented in Figure 5–figure supplement 3 to construct a distance matrix.”

In addition, we changed the title of the Methods section from “Structure determination and refinement of zSmo ectodomain” to “Structure determination, refinement and analyses of the zSmo ectodomain.”

*Are there other uncharacterized receptors that belong to this family*?

We thank the reviewers for this comment. Indeed, CRDs are structurally related distantly to the N-lobe region of the glypican protein family (58; 36; 77). However, the N-lobe does not contain a small molecule binding groove or pocket and many of the Fz/Smo CRD motifs are different (58); thus, we have not included glypicans in our phylogenetic analysis. However, we have now included a note about glypicans in the legend of Figure 5.

*What are the rms deviations for the core parts of the structure, and over how many residues*?

As suggested, we have now included RMSD values and number of equivalent Cα atoms for the structural comparison of the CRDs in the new Figure 5–figure supplement 3.